# Evaluating Unsupervised Denoising Requires Unsupervised Metrics

## Abstract

Unsupervised denoising is a crucial challenge in real-world imaging applications. Unsupervised deep-learning methods have demonstrated impressive performance on benchmarks based on synthetic noise. However, no metrics are available to evaluate these methods in an unsupervised fashion. This is highly problematic for the many practical applications where ground-truth clean images are not available. In this work, we propose two novel metrics: the unsupervised mean squared error (MSE) and the unsupervised peak signal-to-noise ratio (PSNR), which are computed using only noisy data. We provide a theoretical analysis of these metrics, showing that they are asymptotically consistent estimators of the supervised MSE and PSNR. Controlled numerical experiments with synthetic noise confirm that they provide accurate approximations in practice. We validate our approach on real-world data from two imaging modalities: videos in raw format and transmission electron microscopy. Our results demonstrate that the proposed metrics enable unsupervised evaluation of denoising methods based exclusively on noisy data.

## 1 Introduction

Image denoising is a fundamental challenge in image and signal processing, as well as a key preprocessing step for computer vision tasks. Convolutional neural networks achieve state-of-the-art performance for this problem, when trained using databases of clean images corrupted with simulated noise Zhang et al. (2017a). However, in real-world imaging applications such as microscopy, noiseless ground truth videos are often not available. This has motivated the development of unsupervised denoising approaches that can be trained using only noisy measurements Lehtinen et al. (2018); Xie et al. (2020); Laine et al. (2019); Sheth et al. (2021); Huang et al. (2021). These methods have demonstrated impressive performance on natural-image benchmarks, essentially on par with the supervised state of the art. However, to the best of our knowledge, *no unsupervised metrics are currently available to evaluate them using only noisy data*.

Reliance on supervised metrics makes it very challenging to create benchmark datasets using real-world measurements, because obtaining the ground-truth clean images required by these metrics is often either impossible or very constraining. In practice, clean images are typically estimated through temporal averaging, which suppresses dynamic information that is often crucial in scientific applications. Consequently, quantitative evaluation of unsupervised denoising methods is currently almost completely dominated by natural image benchmark datasets with simulated noise Lehtinen et al. (2018); Xie et al. (2020); Laine et al. (2019); Sheth et al. (2021); Huang et al. (2021), which are not always representative of the signal and noise characteristics that arise in real-world imaging applications.

The lack of unsupervised metrics also limits the application of unsupervised denoising techniques in practice. In the absence of quantitative metrics, domain scientists must often rely on visual inspection to evaluate performance on real measurements. This is particularly restrictive for deep-learning approaches, because it makes it impossible to perform systematic hyperparameter optimization and model selection on the data of interest.

In this work, we propose two novel unsupervised metrics to address these issues: the unsupervised mean-squared error (uMSE) and the unsupervised peak signal-to-noise ratio (uPSNR), which are computed *exclusively from noisy data*. These metrics build upon existing unsupervised denoising

methods, which minimize an unsupervised cost function equal to the difference between the denoised estimate and additional noisy copies of the signal of interest Lehtinen et al. (2018). The uMSE is equal to this cost function modified with a correction term, which renders it an unbiased estimator of the supervised MSE.

We provide a theoretical analysis of the uMSE and uPSNR, proving that they are asymptotically consistent estimators of the supervised MSE and PSNR respectively. Controlled experiments on supervised benchmarks, where the true MSE and PSNR can be computed exactly, confirm that the uMSE and uPSNR provide accurate approximations. In addition, we validate the metrics on video data in RAW format, contaminated with real noise that does not follow a known predefined model.

In order to illustrate the potential impact of the proposed metrics on imaging applications where no ground-truth is available, we apply them to transmission-electron-microscopy (TEM) data. Recent advances in direct electron detection systems make it possible for experimentalists to acquire highly time-resolved movies of dynamic events at frame rates in the kilohertz range Faruqi & McMullan (2018); Ercius et al. (2020), which is critical to advance our understanding of functional materials. Acquisition at such high temporal resolution results in severe degradation by shot noise. We show that unsupervised methods based on deep learning can be effective in removing this noise, and that our proposed metrics can be used to evaluate their performance quantitatively using only noisy data.

To summarize, our contributions are (1) two novel unsupervised metrics presented in Section 3, (2) a theoretical analysis providing an asymptotic characterization of their statistical properties (Section 4), (3) experiments showing the accuracy of the metrics in a controlled situation where ground-truth clean images are available (Section 5), (4) validation on real-world videos in RAW format (Section 6), and (5) an application to a real-world electron-microscopy dataset, which illustrates the challenges of unsupervised denoising in scientific imaging (Section 7).

## 2 BACKGROUND AND RELATED WORK

**Unsupervised denoising** The past few years have seen ground-breaking progress in unsupervised denoising, pioneered by Noise2Noise, a technique where a neural network is trained on pairs of noisy images Lehtinen et al. (2018). Our unsupervised metrics are inspired by Noise2Noise, which optimizes a cost function equal to our proposed unsupervised MSE, but without a correction term (which is not needed for training models). Subsequent work focused on performing unsupervised denoising from single images using variations of the *blind-spot* method, where a model is trained to estimate each noisy pixel value using its neighborhood but not the noisy pixel itself (to avoid the trivial identity solution) Krull et al. (2019); Laine et al. (2019); Batson & Royer (2019); Sheth et al. (2021); Xie et al. (2020). More recently, Neighbor2Neighbor revisited the Noise2Noise method, generating noisy image pairs from a single noisy image via spatial subsampling Huang et al. (2021), an insight that can also be leveraged in combination with our proposed metrics, as explained in Section C. Our contribution with respect to these methods is a novel unsupervised metric that can be used for *evaluation*, as it is designed to be an unbiased and consistent estimator of the MSE.

**Stein's unbiased risk estimator (SURE)** provides an asymptotically unbiased estimator of the MSE for i.i.d. Gaussian noise Donoho & Johnstone (1995). This cost function has been used for training unsupervised denoisers Metzler et al. (2018); Soltanayev & Chun (2018); Zhussip et al. (2019); Mohan et al. (2021). In principle, SURE could be used to compute the MSE for evaluation, but it has certain limitations: (1) a closed form expression of the noise likelihood is required, *including the value of the noise parameters* (for example, this is not known for the real-world datasets in Sections 6 and 7), (2) computing SURE requires approximating the divergence of a denoiser (usually via Monte Carlo methods Ramani et al. (2008)), which is computationally very expensive. Developing practical unsupervised metrics based on SURE and studying their theoretical properties is an interesting direction for future research.

**Existing evaluation approaches** In the literature, quantitative evaluation of unsupervised denoising techniques has mostly relied on images and videos corrupted with synthetic noise Lehtinen et al. (2018); Krull et al. (2019); Laine et al. (2019); Batson & Royer (2019); Sheth et al. (2021); Xie et al. (2020). Recently, a few datasets containing real noisy data have been created Abdelhamed et al. (2018); Plotz & Roth (2017); Xu et al. (2018); Zhang et al. (2019). Evaluation on these datasets is based on supervised MSE and PSNR computed from estimated *clean* images obtained by averaging

Figure 1: **MSE vs uMSE.** The traditional supervised mean squared error (MSE) is computed by comparing the denoised estimate to the clean ground truth (left). The proposed unsupervised MSE is computed only from noisy data, via comparison with a noisy reference corresponding to the same ground-truth but corrupted with independent noise (right). A correction term based on two additional noisy references debiases the estimator.

multiple noisy frames. Unfortunately, as a result, the metrics cannot capture dynamically-changing features, which are of interest in many applied domains. In addition, unless the signal-to-noise ratio is quite high, it is necessary to average over a large number of frames to approximate the MSE. For example, as explained in Section E, for CNN based denoiser and an image corrupted by additive Gaussian noise with standard deviation $\sigma = 15$ we need to average $> 1500$ noisy images to achieve the same approximation accuracy as our proposed approach (see Figure 10), which only requires 3 noisy images, and can also be computed from a single noisy image.

**Noise-Level Estimation**. The correction term in uMSE can be interpreted as an estimate of the noise level, obtained by cancelling out the clean signal. In this sense, it is related to noise-level estimation methods Liu et al. (2013); Lebrun et al. (2015); Arias & Morel (2018). However, unlike uMSE, these methods typically assume a parametric model for the noise, and are not used for evaluation.

**No-reference image quality assessment methods** evaluate the perceptual quality of an image Li (2002); Mittal et al. (2012), but not whether it is consistent with an underlying ground-truth corresponding to the observed noisy measurements, which is the goal of our proposed metrics.

## 3 UNSUPERVISED METRICS FOR UNSUPERVISED DENOISING

### 3.1 THE UNSUPERVISED MEAN SQUARED ERROR

The goal of denoising is to estimate a clean signal from noisy measurements. Let $x \in \mathbb{R}^n$ be a signal or a set of signals with $n$ total entries. We denote the corresponding noisy data by $y \in \mathbb{R}^n$. A denoiser $f : \mathbb{R}^n \rightarrow \mathbb{R}^n$ is a function that maps the input $y$ to an estimate of $x$. A common metric to evaluate the quality of a denoiser is the mean squared error between the clean signal and the estimate,

$$\text{MSE} := \frac{1}{n} \sum_{i=1}^{n} \left( x_i - f(y)_i \right)^2. \tag{1}$$

Unfortunately, in most real-world scenarios clean ground-truth signals are not available and evaluation can only be carried out in an *unsupervised* fashion, i.e. *exclusively from the noisy measurements*. In this section we propose an unsupervised estimator of MSE inspired by recent advances in unsupervised denoising Lehtinen et al. (2018). The key idea is to compare the denoised signal to a *noisy reference*, which corresponds to the same clean signal corrupted by independent noise.

In order to motivate our approach, let us assume that the noise is additive, so that $y := x + z$ for a zero-mean noise vector $z \in \mathbb{R}^n$. Imagine that we have access to a noisy reference $a := x + w$ corresponding to the same underlying signal $x$, but corrupted with a different noise realization $w \in \mathbb{R}^n$ independent from $z$ (Section 3.3 explains how to obtain such references in practice). The mean squared difference between the denoised estimate and the reference is approximately equal to

the sum of the MSE and the variance $\sigma^2$ of the noise,

$$\frac{1}{n}\sum_{i=1}^{n}\left(a_i - f(y)_i\right)^2 = \frac{1}{n}\sum_{i=1}^{n}\left(x_i + w_i - f(y)_i\right)^2 \tag{2}$$

$$\approx \frac{1}{n}\sum_{i=1}^{n}\left(x_i - f(y)_i\right)^2 + \frac{1}{n}\sum_{i=1}^{n}w_i^2 \approx \mathrm{MSE} + \sigma^2, \tag{3}$$

because the cross-term $\frac{1}{n}\sum_{i=1}^{n}w_i\left(x_i - f(y)_i\right)^2$ cancels out if $w_i$ and $y_i$ (and hence $f(y_i)$) are independent (and the mean of the noise is zero).

Approximations to equation 3 are used by different unsupervised methods to train neural networks for denoising Lehtinen et al. (2018); Xie et al. (2020); Laine et al. (2019); Huang et al. (2021). The noise term $\frac{1}{n}\sum_{i=1}^{n}w_i^2$ in equation 3 is not problematic for training denoisers as long as it is independent from the input $y$. However, it is definitely problematic for *evaluating* denoisers. In order to estimate the MSE we need to cancel it out. We propose to achieve this by using two other noisy references $b := x + v$ and $c := x + u$, which are noisy measurements corresponding to the clean signal $x$, but corrupted with different, independent noise realizations $v$ and $u$ (just like $a$). Subtracting these references and dividing by two yields an estimate of the noise variance,

$$\frac{1}{n}\sum_{i=1}^{n}\frac{(b_i - c_i)^2}{2} = \frac{1}{n}\sum_{i=1}^{n}\frac{(v_i - u_i)^2}{2} \approx \frac{1}{2n}\sum_{i=1}^{n}v_i^2 + \frac{1}{2n}\sum_{i=1}^{n}u_i^2 \approx \sigma^2, \tag{4}$$

which can then be subtracted from equation 3 to estimate the MSE. This yields our proposed unsupervised metric, which we call unsupervised mean squared error (uMSE), depicted in Figure 1.

**Definition 1** (Unsupervised mean squared error). *Given a noisy input signal $y \in \mathbb{R}^n$ and three noisy references $a$, $b$, $c \in \mathbb{R}^n$ the unsupervised mean squared error of a denoiser $f : \mathbb{R}^n \to \mathbb{R}^n$ is*

$$\mathrm{uMSE} := \frac{1}{n}\sum_{i=1}^{n}\left(a_i - f(y)_i\right)^2 - \frac{(b_i - c_i)^2}{2}. \tag{5}$$

Theorem 4 in Section 4 establishes that the uMSE is a consistent estimator of the MSE as long as (1) the noisy input and the noisy references are independent, (2) their means equal the corresponding entries of the ground-truth clean signal, and (3) their higher-order moments are bounded. These conditions are satisfied by most noise models of interest in signal and image processing, such as Poisson shot noise or additive Gaussian noise. In Section 3.3 we address the question of how to obtain the noisy references required to estimate the uMSE. Section B explains how to compute confidence intervals for the uMSE via bootstrapping.

## 3.2 The Unsupervised Peak Signal-To-Noise Ratio

Peak signal-to-noise ratio (PSNR) is currently the most popular metric to evaluate denoising quality. It is a logarithmic function of MSE defined on a decibel scale,

$$\mathrm{PSNR} := 10\log\left(\frac{\mathrm{M}^2}{\mathrm{MSE}}\right), \tag{6}$$

where $M$ is a fixed constant representing the maximum possible value of the signal of interest, which is usually set equal to 255 for images. Our definition of uMSE can be naturally extended to yield an unsupervised PSNR (uPSNR).

**Definition 2** (Unsupervised peak signal-to-noise ratio). *Given a noisy input signal $y \in \mathbb{R}^n$ and three noisy references $a$, $b$, $c \in \mathbb{R}^n$ the peak signal-to-noise ratio of a denoiser $f : \mathbb{R}^n \to \mathbb{R}^n$ is*

$$\mathrm{uPSNR} := 10\log\left(\frac{\mathrm{M}^2}{\mathrm{uMSE}}\right), \tag{7}$$

*where $M$ is the maximum possible value of the signal of interest.*

Corollary 5 establishes that the uPSNR is a consistent estimator of the PSNR, under the same conditions that guarantee consistency of the uMSE. Section B explains how to compute confidence intervals for the uPSNR via bootstrapping.

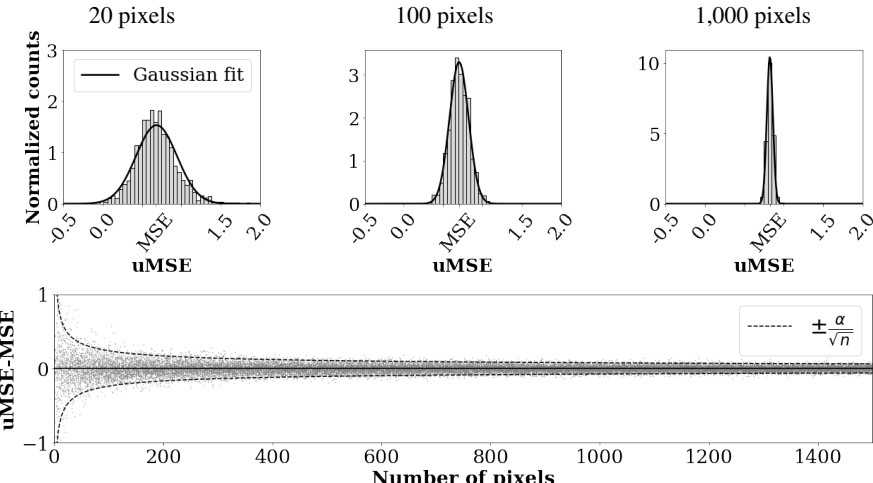

Figure 2: **The uMSE is a consistent estimator of the MSE.** The histograms at the top show the distribution of the uMSE computed from $n$ pixels ($n \in \{20, 100, 1000\}$) of a natural image corrupted with additive Gaussian noise ($\sigma = 55$) and denoised via a deep-learning denoiser (DnCNN). Each point in the histogram corresponds to a different sample of the three noisy references used to compute the uMSE ($\tilde{a}_i$, $\tilde{b}_i$ and $\tilde{c}_i$ in Eq. 9 for $1 \leq i \leq n$), with the same underlying clean pixels. The distributions are centered at the MSE, showing that the estimator is unbiased (Theorem 3), and are well approximated by a Gaussian fit (Theorem 6). As the number of pixels $n$ grows, the standard deviation of the uMSE decreases proportionally to $n^{-1/2}$, and the uMSE converges asymptotically to the MSE (Theorem 4), as depicted in the scatterplot below ($\alpha$ is a constant).

### 3.3 Computing Noisy References In Practice

Our proposed metrics rely on the availability of three noisy references, which ideally should correspond to the same clean image contaminated with independent noise. Deviations between the clean signal in each reference violate Condition 2 in Section 4, and introduce a bias in the metrics. We propose two approaches to compute the references in practice, illustrated in Figure 5.

**Multiple images:** The references can be computed from consecutive frames acquired within a short time interval. This approach is preferable for datasets where the image content does not experience rapid dynamic changes from frame to frame. We apply this approach to the RAW videos in Section 6, where the content is static.

**Single image:** The references can be computed from a single image via spatial subsampling, as described in Section C. Section C shows that this approach is effective as long as the image content is sufficiently smooth with respect to the pixel resolution. We apply this approach to the electron-microscopy data in Section 7, where preserving dynamic content is important.

### 4 Statistical Properties of the Proposed Metrics

In this section, we establish that the proposed unsupervised metrics provide a consistent estimate of the MSE and PSNR. In our analysis, the ground truth signal or set of signals is represented as a deterministic vector $x \in \mathbb{R}^n$. The corresponding noisy data are also modeled as a deterministic vector $y \in \mathbb{R}^n$ that is fed into a denoiser $f : \mathbb{R}^n \rightarrow \mathbb{R}^n$ to produce the denoised estimate $f(y)$. The MSE of the estimate is a deterministic quantity equal to

$$\text{MSE} := \frac{1}{n} \sum_{i=1}^{n} \text{SE}_i, \qquad \text{SE}_i := (x_i - f(y)_i)^2. \tag{8}$$

**Natural images (Gaussian noise)**

Spatial subsampling

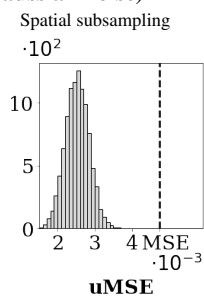

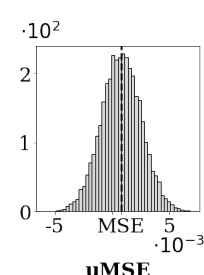

**Electron microscopy (Poisson noise)**

Spatial subsampling

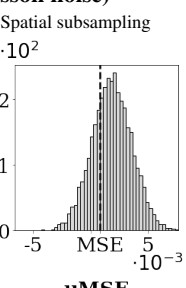

Figure 3: **Bias introduced by spatial subsampling.** The histograms show the distribution of the uMSE (computed as in Figure 2) corresponding to a natural image and a simulated electron-microscopy image corrupted by Gaussian ($\sigma = 55$) and Poisson noise respectively, and denoised with a standard deep-learning denoiser (DnCNN). For each image, the uMSE is computed using noisy references with the same underlying clean image (left), and from noisy references obtained via spatial subsampling (right). For the natural image, spatial subsampling introduces a substantial bias (compare the 1st and 2nd histogram), whereas for the electron-microscopy image the bias is much smaller (compare the 3rd and 4th histogram).

**Noise Model.** The uMSE estimator in Definition 1 depends on three noisy references $\tilde{a}, \tilde{b}, \tilde{c}$, which we model as random variables.[1] Our analysis assumes that these random variables satisfy two conditions:

**Condition 1 (independence):** The entries of $\tilde{a}, \tilde{b}, \tilde{c}$ are all mutually independent.

**Condition 2 (centered noise):** The mean of the $i$th entry of $\tilde{a}, \tilde{b}, \tilde{c}$ equals the corresponding entry of the clean signal, $\mathbb{E}\left[\tilde{a}_i\right] = \mathbb{E}[\tilde{b}_i] = \mathbb{E}\left[\tilde{c}_i\right] = x_i, 1 \leq i \leq n$.

Two popular noise models that satisfy these conditions are:

- *Additive Gaussian*, where $\tilde{a}_i := x_i + \tilde{w}_i$, $\tilde{b}_i := x_i + \tilde{v}_i$, $\tilde{c}_i := x_i + \tilde{u}_i$, for i.i.d. Gaussian $\tilde{w}_i, \tilde{v}_i, \tilde{u}_i$.

- *Poisson*, where $\tilde{a}_i, \tilde{b}_i, \tilde{c}_i$ are i.i.d. Poisson random variables with parameter $x_i$.

**Theoretical Guarantees.** Our goal is to study the statistical properties of the uMSE

$$\widetilde{\text{uMSE}} := \frac{1}{n}\sum_{i=1}^{n}\widetilde{\text{uSE}}_i, \qquad \widetilde{\text{uSE}}_i := (\tilde{a}_i - f(y)_i)^2 - \frac{(\tilde{b}_i - \tilde{c}_i)^2}{2}. \tag{9}$$

As indicated by the tilde, under our modeling assumptions, the uMSE is a random variable. We first show that the correction factor in the definition of uMSE succeeds in debiasing the estimator, so that its mean is equal to the MSE.

**Theorem 3** (The uMSE is unbiased, proof in Section G.1)**.** *If Conditions 1 and 2 hold, the uMSE is an unbiased estimator of the MSE, i.e.* $\mathbb{E}[\widetilde{\text{uMSE}}] = \text{MSE}$.

Theorem 3 establishes that the distribution of the uMSE is centered at the MSE. We now show that its variance shrinks at a rate inversely proportional to the number of signal entries $n$, and therefore converges to the MSE in mean square and probability as $n \rightarrow \infty$ (see Figure 2 for a numerical demonstration). This occurs as long as the higher central moments of noise and the entrywise denoising error are bounded by a constant, which is to be expected in most realistic scenarios.

**Theorem 4** (The uMSE is consistent, proof in Section G.2)**.** *Let* $\mu_i^{[k]}$ *denote the $k$th central moment of* $\tilde{a}_i, \tilde{b}_i, \tilde{c}_i$*, and* $\gamma := \max_{1 \leq i \leq n} |x_i - f(y)_i|$ *the maximum entrywise denoising error. If Conditions 1 and 2 hold, and there exists a constant $\alpha$ such that* $\max_{1 \leq i \leq n} \max\left\{\mu_i^{[4]}, \mu_i^{[3]}\gamma, \gamma^4\right\} \leq \alpha$*, then the*

---

[1]In our analysis, all random quantities are marked with a tilde for clarity.

Table 1: **Controlled comparison of PSNR and uPSNR.** The table shows the PSNR computed from clean ground-truth images, compared to two versions of the proposed estimator: one using noisy references corresponding to the same clean image (uPSNR), and another using a single noisy image combined with spatial subsampling (uPSNR$_s$). The metrics are compared on the datasets and denoising methods described in Section H

.

**Natural images (Gaussian noise)**

| Method | $\sigma = 25$ | | | $\sigma = 50$ | | | $\sigma = 75$ | | | $\sigma = 100$ | | |
|---|---|---|---|---|---|---|---|---|---|---|---|---|
| | PSNR | uPSNR | uPSNR$_S$ | PSNR | uPSNR | uPSNR$_S$ | PSNR | uPSNR | uPSNR$_S$ | PSNR | uPSNR | uPSNR$_S$ |
| Bilateral | 24.20 | 24.18 | 26.20 | 21.84 | 21.86 | 22.90 | 19.14 | 19.17 | 19.58 | 16.30 | 16.37 | 16.47 |
| DenseNet | 26.54 | 26.51 | 27.61 | 23.98 | 24.06 | 26.28 | 22.75 | 23.00 | 24.69 | 21.92 | 21.97 | 23.78 |
| DnCNN | 26.19 | 26.21 | 28.14 | 23.95 | 24.02 | 26.08 | 22.72 | 22.75 | 24.59 | 21.84 | 21.84 | 23.71 |
| UNet | 26.29 | 26.28 | 27.98 | 23.92 | 24.01 | 26.25 | 22.68 | 22.70 | 24.84 | 21.91 | 21.89 | 23.48 |

**Electron microscopy (Poisson noise)**

| Bilateral | | | BlindSpot | | | DnCNN | | | UNet | | |
|---|---|---|---|---|---|---|---|---|---|---|---|
| PSNR | uPSNR | uPSNR$_S$ | PSNR | uPSNR | uPSNR$_S$ | PSNR | uPSNR | uPSNR$_S$ | PSNR | uPSNR | uPSNR$_S$ |
| 20.18 | 20.20 | 20.21 | 24.86 | 24.87 | 24.74 | 25.74 | 25.68 | 25.86 | 24.65 | 24.69 | 24.79 |

*mean squared error between the MSE and the uMSE satisfies the bound*

$$\mathbb{E}\left[\left(\widetilde{\mathrm{uMSE}} - \mathrm{MSE}\right)^2\right] = \textcolor{red}{\mathrm{Var}\left[\widetilde{\mathrm{uMSE}}\right]} \leq \frac{\alpha}{n}. \tag{10}$$

*Consequently, $\lim_{n \to \infty} \mathbb{E}[(\widetilde{\mathrm{uMSE}} - \mathrm{MSE})^2] = 0$, so the uMSE converges to the MSE in mean square and therefore also in probability.*

Consistency of the uMSE implies consistency of the uPSNR.

**Corollary 5** (The uPSNR is consistent, proof in Section G.3). *Under the assumptions of Theorem 4, the uPSNR defined as*

$$\widetilde{\mathrm{uPSNR}} := 10 \log\left(\frac{\mathrm{M}^2}{\widetilde{\mathrm{uMSE}}}\right), \tag{11}$$

*where $M$ is a fixed constant, converges in probability to the PSNR, as $n \to \infty$.*

The uMSE converges to a Gaussian random variable asymptotically as $n \to \infty$.

**Theorem 6** (The uMSE is asymptotically normally distributed, proof in Section G.4). *If the first six central moments of $\tilde{a}_i$, $\tilde{b}_i$, $\tilde{c}_i$ and the maximum entrywise denoising error $\max_{1 \leq i \leq n} |x_i - f(y)_i|$ are bounded, and Conditions 1 and 2 hold, the uMSE is asymptotically normally distributed as $n \to \infty$.*

Our numerical experiments show that the distribution of the uMSE is well approximated as Gaussian even for relatively small values of $n$ (see Figure 2). This can be exploited to build confidence intervals for the uMSE and uPSNR, as explained in Section B.

## 5 CONTROLLED EVALUATION OF THE PROPOSED METRICS

In this section, we study the properties of the uMSE and uPSNR through numerical experiments in a controlled scenario where the ground-truth clean images are known. We use a dataset of natural images (Martin et al., 2001; Zhang et al., 2017b; Franzen, 1993) corrupted with additive Gaussian noise, and a dataset of simulated electron-microscopy images Vincent et al. (2021) corrupted with Poisson noise. For the two datasets, we compute the supervised MSE and PSNR using the ground-truth clean image. To compute the uMSE and uPSNR we use noisy references corresponding to the same clean image corrupted with independent noise. We also compute the uMSE and uPSNR using noisy references obtained from a single noisy image via spatial subsampling, as described in

Table 2: **Comparison of averaging-based PSNR and uPSNR on RAW videos with real noise.** The proposed uPSNR metric, computed using three noisy references, is very similar to an averaging-based PSNR estimate computed from 10 noisy references. The metrics are compared on the datasets and denoising methods described in Section I.

| ISO | Image (Wavelet) | | Image (CNN) | | Video (Temp. Avg) | | Video (CNN) | |
|---|---|---|---|---|---|---|---|---|
| | $PSNR_{avg}$ | uPSNR | $PSNR_{avg}$ | uPSNR | $PSNR_{avg}$ | uPSNR | $PSNR_{avg}$ | uPSNR |
| 1600 | 37.56 | 37.76 | 46.88 | 48.05 | 34.32 | 34.36 | 48.06 | 49.51 |
| 3200 | 35.52 | 35.55 | 44.91 | 45.51 | 32.48 | 32.47 | 46.45 | 47.33 |
| 6400 | 32.60 | 32.68 | 42.74 | 43.05 | 30.77 | 30.75 | 44.75 | 45.16 |
| 12800 | 28.43 | 28.46 | 40.22 | 39.75 | 27.71 | 27.76 | 42.22 | 41.69 |
| 25600 | 26.79 | 26.9 | 40.19 | 38.78 | 27.08 | 27.12 | 42.13 | 40.32 |
| Mean | 32.18 | 32.27 | 42.99 | 43.03 | 30.47 | 30.49 | 44.72 | 44.80 |

Section C, which we denote by $uMSE_S$ and $uPSNR_S$ respectively.[2] All metrics are applied to multiple denoising approches, as described in more detail in Section H.

The results are reported in Tables 1 and 3. When the noisy references correspond exactly to the same clean image (and therefore satisfy the conditions in Section 4), the unsupervised metrics are extremely accurate across different noise levels for all denoising methods.

**Single-image results:** When the noisy references are computed via spatial subsampling, the metrics are still very accurate for the electron-microscopy dataset, but less so for the natural-image dataset if the PSNR is high (above 20 dB). The culprit is the difference between the clean images underlying each noisy reference (see Figure 5), which introduces a bias in the unsupervised metric, depicted in Figure 3. Figure 8 shows that the difference is more pronounced in natural images than in electron-microscopy images, which are smoother with respect to the pixel resolution (see Section C).

## 6 APPLICATION TO VIDEOS IN RAW FORMAT

We evaluate our proposed metrics on a dataset of videos in raw format, consisting of direct readings from the sensor of a surveillance camera contaminated with real noise at five different ISO levels Yue et al. (2020). The dataset contains 11 unique videos divided into 7 segments, each consisting of 10 noisy frames that capture the same static object. We consider four different denoisers: a wavelet-based method, temporal averaging and two versions of a state-of-the-art unsupervised deep-learning method using images and videos respectively Sheth et al. (2021). A detailed description of the experiments is provided in Section I. Tables 4 and 2 compare our proposed unsupervised metrics (computed using three noisy frames in each segment) with MSE and PSNR estimates obtained via averaging from ten noisy frames. The two types of metric yield similar results: the deep-learning methods clearly outperform the other baselines, and the video-based methods outperform the image-based methods. As explained in Section E, the averaging-based MSE and PSNR are not consistent estimators of the true MSE and PSNR, and can be substantially less accurate than the uMSE and uPSNR (see Figure 10), so they should not be considered ground-truth metrics.

## 7 APPLICATION TO ELECTRON MICROSCOPY

Our proposed metrics enable quantitative evaluation of denoisers in the absence of ground-truth clean images. We showcase this for transmission electron microscopy (TEM), a key imaging modality in material sciences. Recent developments enable the acquisition of high frame-rate images, capturing high temporal resolution dynamics, thought to be crucial in catalytic processes Crozier et al. (2019). Images acquired under these conditions are severely limited by noise. Recent work suggest that deep learning methods provide an effective solution Sheth et al. (2021); Mohan et al. (2022; 2021), but, instead of quantitative metrics, evaluation on real data has been limited to visual inspection.

---

[2]The other metrics are applied to subsampled images in order to make them directly comparable to the $uMSE_S$ and $uPSNR_S$.

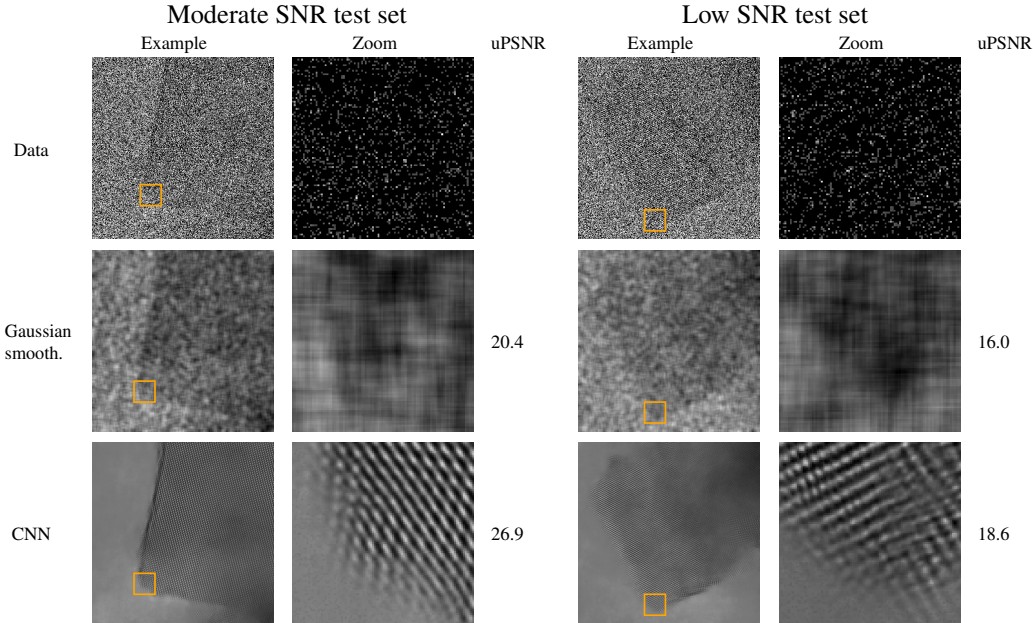

Figure 4: **Denoising real-world electron-microscopy data.** Example noisy images (top) from the moderate-SNR (left 2 columns) and low-SNR (right 2 columns) test sets described in Section 7. The data are denoised using a Gaussian-smoothing baseline (second row) and a convolutional neural network (CNN). The uPSNR of each method on each test set is shown beside the images. Both the uPSNR and visual inspection indicate that the CNN clearly outperforms Gaussian smoothing, and that both methods achieve much worse results on the low-SNR test set.

The TEM dataset consists of 18,597 noisy frames depicting platinum nanoparticles on a cerium oxide support. A major challenge for the application of unsupervised metrics is the presence of local correlations in the noise (see Figure 13). We address this by performing spatial subsampling to reduce the correlation and selecting two contiguous test sets with low correlation: 155 images with *moderate* signal-to-noise ratio (SNR), and 383 images with *low* SNR, which are more challenging. We train an unsupervised Neighbor2Neighbor convolutional-neural network Huang et al. (2021) on a training set containing 70% of the data, and compare its performance to a Gaussian-smoothing baseline on the two test sets. Section J provides a more detailed description of the dataset and the models.

Figure 4 shows examples from the data and the corresponding denoised images, as well as the uPSNR of each method for the two test sets. Figure 11 shows a histogram of the uMSE values for each individual test image. The unsupervised metrics indicate that the deep-learning method achieves effective denoising on the moderate SNR set, clearly outperforming the Gaussian-smoothing baseline, and also that both methods produce significantly worse results for the low-SNR test set. Figure 12 shows that uMSE produces consistent image-level evaluations for the two denoisers. These conclusions are supported by the visual appearance of the images.

## 8 CONCLUSION AND OPEN QUESTIONS

In this work we introduce two novel unsupervised metrics computed exclusively from noisy data, which are asymptotically consistent estimators of the corresponding supervised metrics, and yield accurate approximations in practice. These results are limited to denoising under the assumption of independent noise. Several important open questions remain: (1) How to address the bias introduced by spatial subsampling and achieve an unbiased approximation to the MSE from a single noisy image. (2) How to account for noise distributions and artifacts which are not pixel-wise independent (see e.g. Prakash et al. (2021)). (3) How to obtain unsupervised approximations of perceptual metrics such as SSIM Wang et al. (2004). (4) How to perform unsupervised evaluation for general inverse problems, and related applications such as realistic image synthesis Zwicker et al. (2015).

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

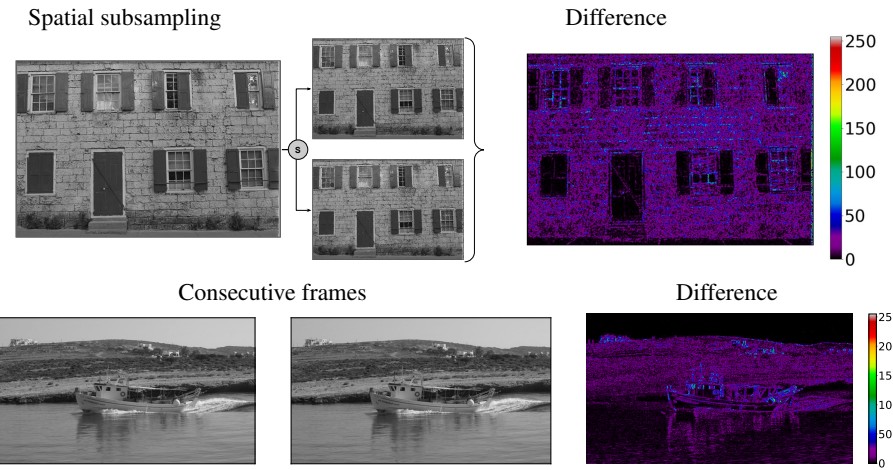

Figure 5: **Noisy references.** The proposed metrics require noisy references corresponding to the same clean image corrupted by independent noise. These references can be obtained from a single image via spatial subsampling (above) or from consecutive frames (below). In both cases, there may be small differences in the signal content of the references, shown by the corresponding heatmaps.

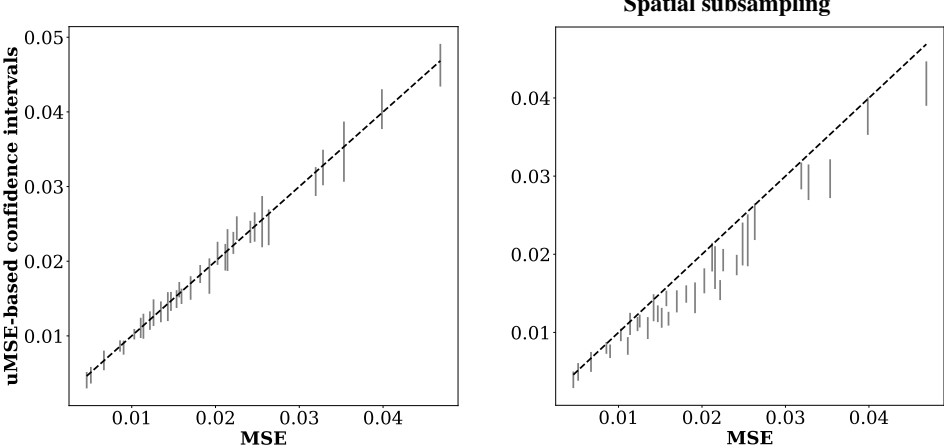

Figure 6: **Unsupervised confidence intervals for the MSE.** 0.95-Confidence intervals computed following Algorithm 7 of natural images from the dataset in Section 5 corrupted with additive Gaussian noise ($\sigma = 55$) and denoised via a standard deep-learning denoiser (DnCNN). The horizontal coordinate of each interval corresponds to the true MSE, so ideally 95% of the intervals should overlap with the diagonal dashed identity line. The left plot shows that this is the case when the noisy references with the same underlying clean image, demonstrating that Algorithm 7 produces valid confidence intervals. The right plot shows confidence intervals based on noisy references obtained via spatial subsampling (right). Spatial subsampling produces a systematic bias in the uMSE, analyzed in Section C which shifts the intervals away from the identity line when the underlying image content is not sufficiently smooth with respect to the pixel resolution.

## A  COMPUTATION OF NOISY REFERENCES

Figure 5 illustrates two approaches to compute the noisy references required by the proposed metrics, which are described in Section 3.3.

# B  CONFIDENCE INTERVALS FOR UNCERTAINTY QUANTIFICATION

The uMSE and uPSNR are estimates of the MSE and PSNR computed from noisy data, so they are inherently uncertain. We propose to quantify this uncertainty using confidence intervals obtained via bootstrapping.

**Algorithm 7** (Bootstrap confidence intervals). *We assume access to a noisy input signal $y \in \mathbb{R}^n$ and three noisy references $a$, $b$, $c \in \mathbb{R}^n$. For $1 \leq k \leq K$, build an index set $\mathcal{B}_k$ by sampling $n$ entries from $\{1, 2, \ldots, n\}$ uniformly and independently at random with replacement. Then set*

$$\text{uMSE}_k := \frac{1}{n} \sum_{i \in \mathcal{B}_k} (a_i - f(y)_i)^2 - \frac{(b_i - c_i)^2}{2}, \qquad \text{uPSNR}_k := 10 \log \left( \frac{\text{M}^2}{\text{uMSE}_k} \right). \quad (12)$$

*To build $1 - \alpha$ confidence intervals, $0 < \alpha < 1$ for the uMSE and uPSNR set*

$$\mathcal{I}_{\text{uMSE}} := \left[ q^{\text{uMSE}}_{\alpha/2}, q^{\text{uMSE}}_{1-\alpha/2} \right], \qquad \mathcal{I}_{\text{uPSNR}} := \left[ q^{\text{uPSNR}}_{\alpha/2}, q^{\text{uPSNR}}_{1-\alpha/2} \right], \quad (13)$$

*where $q^{\text{uMSE}}_{\alpha/2}$ and $q^{\text{uMSE}}_{1-\alpha/2}$ are the $\alpha/2$ and $1 - \alpha/2$ quantiles of the set $\{\text{uMSE}_1, \ldots, \text{uMSE}_K\}$, and $q^{\text{uPSNR}}_{\alpha/2}$ and $q^{\text{uPSNR}}_{1-\alpha/2}$ are the $\alpha/2$ and $1 - \alpha/2$ quantiles of the set $\{\text{uPSNR}_1, \ldots, \text{uPSNR}_K\}$.*

Theorem 6 establishes that the uMSE is asymptotically normal. In addition, our numerical experiments show that the distribution of the uMSE is well approximated as Gaussian even for relatively small values of $n$ (see Figure 2). As a result, the bootstrap confidence intervals for the uMSE produced by Algorithm 7 contain the MSE with probability approximately $1 - \alpha$ (see Section 13.3 in Efron & Tibshirani (1994)). This also implies that the PSNR belongs to the bootstrap confidence intervals for the uPSNR with probability approximately $1 - \alpha$ because the function that maps the uMSE to the uPSNR and the MSE to the PSNR is monotone (see Section 13.6 in Efron & Tibshirani (1994)).

Figure 6 shows a numerical verification that the proposed approach yields valid confidence intervals for MSE in the controlled experiments of Section 5, where the ground-truth clean images are known. It also shows that the bias introduced by spatial subsampling for natural images (see Section C), shifts the confidence intervals away from the true MSE.

# C  SPATIAL SUBSAMPLING

In this section, we propose a method to obtain the noisy references required to estimate uMSE and uPSNR. We focus our discussion on images, but similar ideas can be applied to videos and time-series data. In order to simplify the notation, we consider $N \times N$ images. The $n$-dimensional signals in other sections can be interpreted as vectorized versions of these images with $n = N^2$.

We assume that we have available a noisy image $I$ of dimensions $2N \times 2N$. We extract four noisy references from $I$ by spatial subsampling. The method is inspired by the Neighbor2Neighbor unsupervised denoising method, which uses random subsampling to generate noisy image pairs during training Huang et al. (2021). Figure 7 illustrates the approach.

**Algorithm 8** (Decomposition via spatial subsampling). *Given an image $I \in \mathbb{R}^{2N \times 2N}$, let*

$$
\begin{aligned}
S_1(i, j) &:= I(2i-1, 2j-1), & S_2(i, j) &:= I(2i, 2j-1), \\
S_3(i, j) &:= I(2i-1, 2j), & S_4(i, j) &:= I(2i, 2j), & 1 \leq i, j \leq n. \quad (14)
\end{aligned}
$$

*The spatial decomposition of $I$ is equal to four sub-images $Y$, $A$, $B$, $C \in \mathbb{R}^{N \times N}$ where $Y(i, j)$, $A(i, j)$, $B(i, j)$, $C(i, j)$ are set equal to $S_1(i, j)$, $S_2(i, j)$, $S_3(i, j)$, $S_4(i, j)$, or to a random permutation of the four values.*

# D  EFFECT OF SPATIAL SUBSAMPLING ON THE PROPOSED METRICS

Spatial subsampling generates four noisy sub-images that correspond to the noisy input $y$ and the three noisy references $a$, $b$ and $c$ in Definition 1. In our derivation of the uMSE, we assume that these

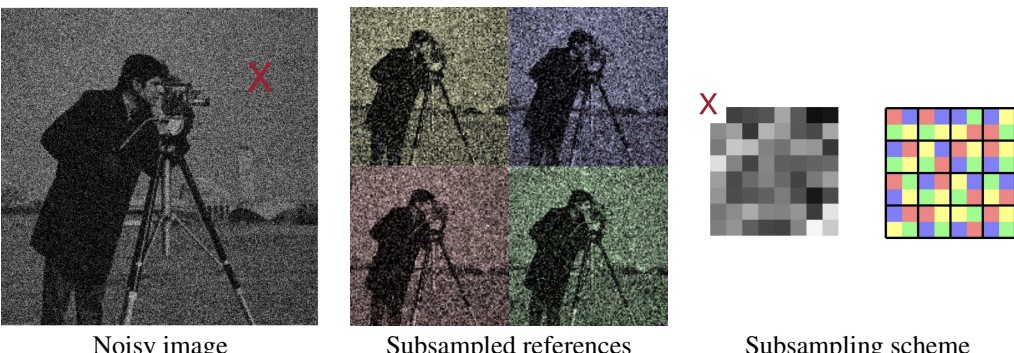

Noisy image      Subsampled references      Subsampling scheme

Figure 7: **Spatial subsampling** uses a single noisy image (left) to extract four noisy references (center) corresponding approximately to the same underlying clean image, but with independent noise. The pixels of each $2 \times 2$ block are assigned to each of the references either deterministically, or at random (right).

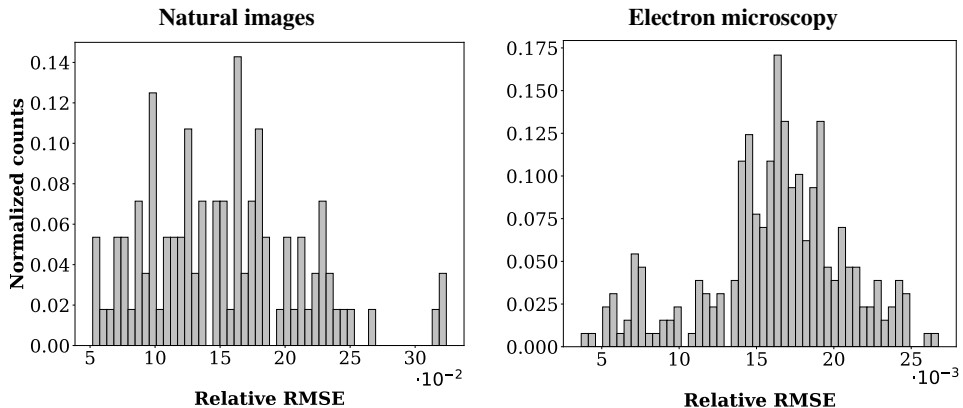

Figure 8: **Effect of spatial subsampling.** The histograms show the relative root mean square error (RMSE) between clean copies of images obtained via spatial subsampling following Algorithm 8 for the natural images (left) and electron-microscopy images (right) used for the experiments in Section 5. The difference is substantially larger in natural images, because they are less smooth with respect to the pixel resolution than the electron-microscopy images.

four noisy signals are generated by corrupting the same ground-truth clean signal with independent noise. This holds for the sub-images in Definition 8 if (1) the underlying clean image is smooth, so that adjacent pixels are approximately equal, and (2) the noise is pixel-wise independent. Tables 1 and 3, and Figures 3 and 6 show that these assumptions don't hold completely for natural images, which introduces a bias in the uMSE. This bias also exists for the electron-microscopy images but it is much smaller, because the images are smoother with respect to the pixel resolution. Figure 8 shows the relative root mean square error (RMSE) between clean copies of images obtained via spatial subsampling following Algorithm 8 for the natural images (left) and electron-microscopy images (right) used for the experiments in Section 5. The difference is substantially larger in natural images, because they are less smooth with respect to the pixel resolution than the electron-microscopy images.

In order to further analyze the effect of spatial subsampling on the proposed metrics, we performed a controlled experiment where we applied different degrees of smoothing (via a Gaussian filter) to a natural image. We evaluated the relative RMSE of the corresponding subsampled references. In addition, we fed the smoothed images contaminated by noise into a denoiser and compared the uMSE of the denoised image with its true MSE. The results are shown in Figure 9. We observe that smoothing results in a stark decrease of both the relative RMSE and the uMSE, suggesting that spatial

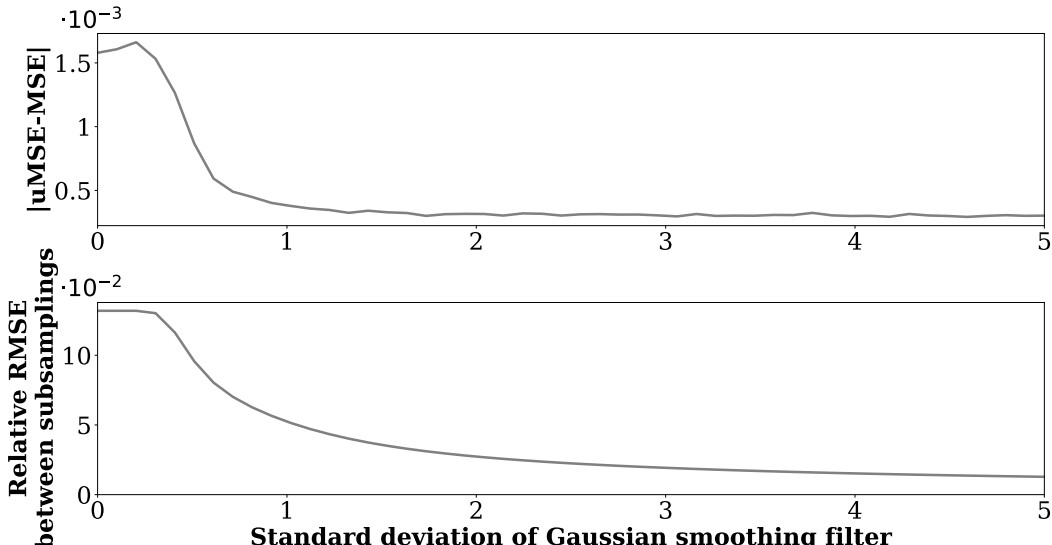

Figure 9: **The bias produced by spatial subsampling is related to image smoothness.** The top graph shows the relative RMSE of the subsampled references corresponding to a natural image (the same as in Figure 3), after smoothing with a Gaussian filter with different standard deviations. In order to evaluate the effect of image smoothness on the uMSE, we fed the smoothed images contaminated by Gaussian i.i.d. noise with standard deviation equal to 55 into a DnCNN denoiser (as in Figure 3). The bottom graph shows the absolute difference between the MSE and the uMSE as a function of the smoothness of the underlying clean image. Smoothing results in a clear decrease of both the relative RMSE and the uMSE, suggesting that spatial subsampling is effective as long as the underlying image content is sufficiently smooth with respect to the pixel resolution.

subsampling is effective as long as the underlying image content is sufficiently smooth with respect to the pixel resolution (as supported also by the results on the electron-microscopy data).

## E    COMPARISON WITH AVERAGING-BASED MSE ESTIMATION

Existing denoising benchmarks containing images corrupted with real noise perform evaluation by computing the MSE or PSNR using an estimate of the *clean* image obtained by averaging multiple noisy frames Abdelhamed et al. (2018); Plotz & Roth (2017); Xu et al. (2018); Zhang et al. (2019). In this section, we show both theoretically and numerically that this approach produces a poor estimate of the MSE and PSNR, unless the signal-to-noise ratio of the data is very low, or we use a large number of noisy frames.

The following lemma shows that in contrast to our proposed metric uMSE, the approximation to the MSE obtained via averaging is biased and *not consistent*, in the sense that it does not converge to the true MSE when the number of pixels tends to infinity. The metric does converge to the MSE as the number of noisy images tends to infinity, but this is of little practical significance, since this number cannot be increased arbitrarily in actual applications.

**Lemma 9** (MSE via averaging). *Consider a clean signal $x \in \mathbb{R}^n$, an estimate $f(y) \in \mathbb{R}^n$ (obtained by applying a denoiser $f$ to the data $y \in \mathbb{R}^n$), and $m$ noisy references*

$$\tilde{r}_i^{[m]} := x_i + \tilde{z}_i^{[m]}, \quad 1 \le i \le n, 1 \le j \le m, \tag{15}$$

*where $\tilde{z}_i^{[m]}$, $1 \le i \le n, 1 \le j \le m$, are i.i.d. zero-mean Gaussian random variables with variance $\sigma^2$. We define the averaging-based MSE as*

$$MSE_{avg} := \frac{1}{n} \sum_{i=1}^{n} \left( \frac{1}{m} \sum_{j=1}^{m} \tilde{r}_i^{[m]} - f(y)_i \right)^2. \tag{16}$$

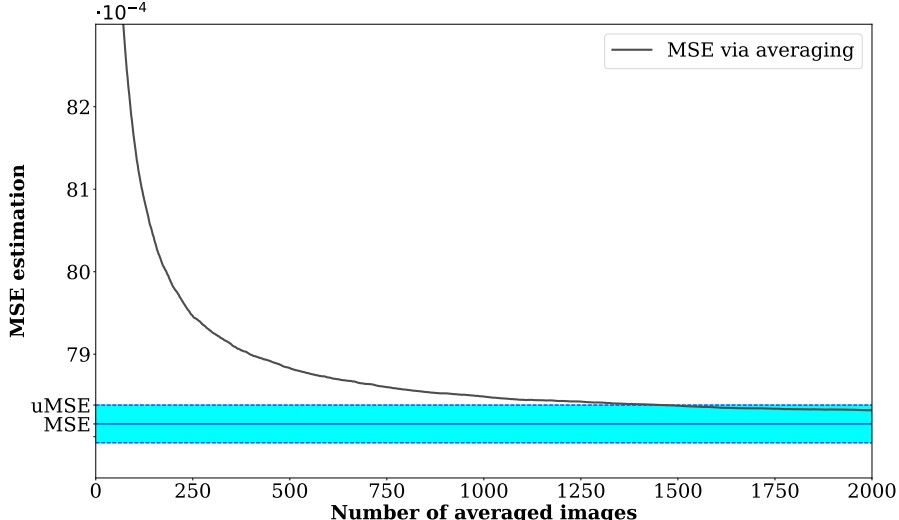

Figure 10: **Comparison of averaging-based MSE and uMSE.** The plot shows the MSE, uMSE and averaging-based $MSE_{avg}$ corresponding to a natural image corrupted by Gaussian ($\sigma = 15$, to simulate a noise level similar to that of the RAW videos in Section 6) and denoised with a standard deep-learning denoiser (DnCNN). The uMSE is computed with 3 noisy references. The averaging-based $MSE_{avg}$ is computed with different number of noisy references indicated by the horizontal axis. The blue shaded region corresponds to an error that is smaller or equal to the error incurred by the uMSE. Averaging-based MSE requires 1,510 noisy images to achieve this accuracy.

*The $MSE_{avg}$ is a biased estimator of the true MSE*

$$MSE := \frac{1}{n} \sum_{i=1}^{n} (x_i - f(y)_i)^2,\tag{17}$$

*since its mean equals*

$$\mathbb{E}\left[MSE_{avg}\right] = MSE + \frac{\sigma^2}{m}.\tag{18}$$

*Proof.* By the assumptions, and linearity of expectation,

$$\mathbb{E}\left[\mathrm{MSE}_{avg}\right] = \mathbb{E}\left[\frac{1}{n} \sum_{i=1}^{n} \left(\frac{1}{m} \sum_{j=1}^{m} \tilde{z}_i^{[m]} + x_i - f(y)_i\right)^2\right]\tag{19}$$

$$= \frac{1}{n} \sum_{i=1}^{n} (x_i - f(y)_i)^2 + \frac{1}{n} \sum_{i=1}^{n} \mathbb{E}\left[\left(\frac{1}{m} \sum_{j=1}^{m} \tilde{z}_i^{[m]}\right)^2\right]\tag{20}$$

$$= \mathrm{MSE} + \frac{\sigma^2}{m}.\tag{21}$$

$\square$

As established in Section 4, the proposed uMSE metric is an unbiased estimator of the MSE that is consistent as $n \to \infty$ and only requires $m := 3$ noisy references. Figure 10 shows a numerical comparison between uMSE and the averaging-based MSE for one of the natural images used in the experiments of Section 5. We observe that averaging-based MSE requires $m := 1510$ in order to match the accuracy achieved by the uMSE with only three noisy references.

Table 3: **Controlled comparison of MSE and uMSE.** The table shows the MSE computed from clean ground-truth images, compared to two versions of the proposed estimator: one using noisy references corresponding to the same clean image (uMSE), and another using a single noisy image combined with spatial subsampling (uMSE$_s$). The metrics are compared on the datasets and denoising methods described in Section H

.

**Natural images (Gaussian noise)** $\cdot 10^{-3}$

| Method | $\sigma = 25$ | | | $\sigma = 50$ | | | $\sigma = 75$ | | | $\sigma = 100$ | | |
|---|---|---|---|---|---|---|---|---|---|---|---|---|
| | MSE | uMSE | uMSE$_S$ | MSE | uMSE | uMSE$_S$ | MSE | uMSE | uMSE$_S$ | MSE | uMSE | uMSE$_S$ |
| Bilateral | 4.38 | 4.4 | 2.64 | 6.88 | 6.87 | 5.26 | 12.3 | 12.3 | 11.1 | 23.5 | 23.2 | 22.5 |
| DenseNet | 2.58 | 2.59 | 2.11 | 4.70 | 4.65 | 2.81 | 6.23 | 6.16 | 4.02 | 7.44 | 7.44 | 5.00 |
| DnCNN | 2.85 | 2.84 | 1.81 | 4.72 | 4.71 | 2.85 | 6.21 | 6.24 | 3.96 | 7.48 | 7.60 | 5.05 |
| UNet | 2.76 | 2.77 | 1.91 | 4.78 | 4.76 | 2.84 | 6.32 | 6.22 | 3.89 | 7.47 | 7.68 | 5.05 |

**Electron microscopy (Poisson noise)** $\cdot 10^{-3}$

| Bilateral | | | BlindSpot | | | DnCNN | | | UNet | | |
|---|---|---|---|---|---|---|---|---|---|---|---|
| MSE | uMSE | uMSE$_S$ | MSE | uMSE | uMSE$_S$ | MSE | uMSE | uMSE$_S$ | MSE | uMSE | uMSE$_S$ |
| 9.57 | 9.55 | 9.54 | 3.97 | 4.00 | 3.96 | 3.00 | 3.03 | 2.94 | 4.18 | 4.10 | 4.12 |

Table 4: **Comparison of averaging-based MSE and uMSE on RAW videos with real noise.** The proposed uMSE metric, computed using three noisy references, is very similar to an averaging-based PSNR estimate computed from 10 noisy references. The metrics are compared on the datasets and denoising methods described in Section I. All numbers in the table are scaled by $\cdot 10^{-4}$

| ISO | Image (Wavelet) | | Image (CNN) | | Video (Temp. Avg) | | Video (CNN) | |
|---|---|---|---|---|---|---|---|---|
| | MSE$_{avg}$ | uMSE | MSE$_{avg}$ | uMSE | MSE$_{avg}$ | uMSE | MSE$_{avg}$ | uMSE |
| 1600 | 1.795 | 1.69 | 0.284 | 0.183 | 5.317 | 5.282 | 0.234 | 0.131 |
| 3200 | 2.887 | 2.866 | 0.379 | 0.336 | 8.151 | 8.134 | 0.267 | 0.217 |
| 6400 | 5.792 | 5.702 | 0.686 | 0.624 | 10.503 | 10.573 | 0.433 | 0.361 |
| 12800 | 15.205 | 15.11 | 1.22 | 1.31 | 19.967 | 19.871 | 0.741 | 0.78 |
| 25600 | 23.194 | 22.549 | 1.63 | 1.737 | 21.277 | 21.1 | 1.052 | 1.079 |
| Mean | 9.775 | 9.583 | 0.84 | 0.838 | 13.043 | 12.992 | 0.545 | 0.514 |

## F  ADDITIONAL RESULTS

This section contains additional results, which are not included in the main paper due to space constraints. They include:

- Table 3 shows a controlled comparison of MSE and uMSE on clean ground-truth images, and noisy frames for both natural images with additive Gaussian noise and TEM images with Poisson noise.
- Table 4 shows a comparison between averaging-based MSE and uMSE on RAW videos with real noise described in Sections 6 and I.
- Figure 11 Shows the estimates of uMSE evaluated on TEM data for two denoisers, described in Section 7.
- Figure 12 Shows that uMSE provides a consistent ordering of images that are easier/harder to denoise, across different denoisers
- Figure 13 shows the empirical correlation of neighboring pixels in the TEM data, necessitating subsampling in order to evaluate uMSE, uPSNR.

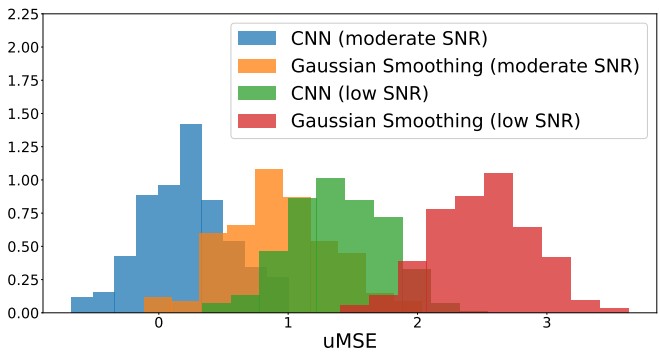

Figure 11: **uMSE for real-world electron-microscopy data.** The figure shows the histograms of the uMSE (computed from a single noisy image via spatial subsampling) of two denoisers (CNN and Gaussian smoothing kernel) on the two test sets described in Section 7. The uMSE discriminates between the different methods and test sets, in a way that is consistent with the visual appearance of the denoised images.

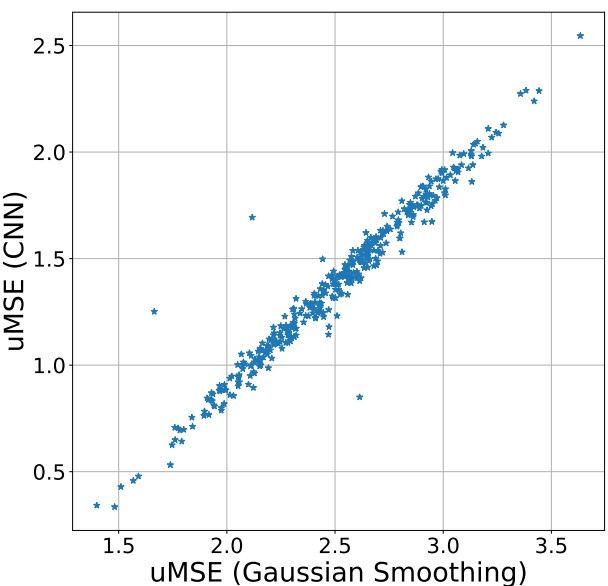

Figure 12: **uMSE produces consistent image-level evaluations across different denoisers.** We compare the uMSE estimate per image for both the CNN and the Gaussian smoothing denoisers on the low SNR data (green and red histograms in Figure 11). While the ranges are different for each denoiser (the CNN denoises more effectively), the uMSE values are highly correlated, indicating that uMSE provides a consistent evaluation of the individual images.

## G  PROOFS

### G.1  PROOF OF THEOREM 3

The following lemma shows that each individual term in the uMSE is unbiased.

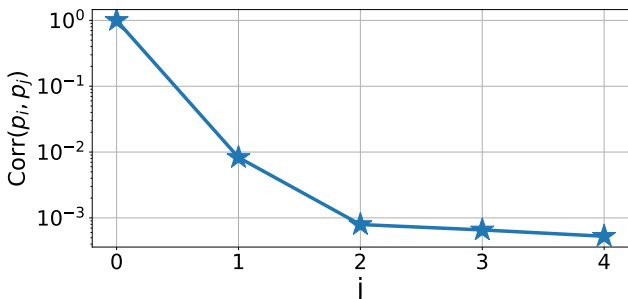

Figure 13: **Empirical correlation of adjacent pixels in electron-microscopy data.** The graph shows the correlation coefficient between a pixel and a pixel that is $j$ pixels away for different values of $j$. The correlation coefficient is computed after subtracting a mean computed via averaging across frames. The correlation between adjacent pixels is particularly high, so spatial subsampling by a factor of two substantially reduces the pixel-wise correlation.

**Lemma 10** (Proof in Section G.5.2). *If Conditions 1 and 2 in Section 4 hold,*

$$\mathbb{E}\left[\widetilde{\mathrm{uSE}}_i\right] = \mathrm{SE}_i, \qquad 1 \leq i \leq n. \tag{22}$$

The proof then follows immediately from linearity of expectation,

$$\mathbb{E}\left[\widetilde{\mathrm{uMSE}}\right] = \mathbb{E}\left[\frac{1}{n}\sum_{i=1}^{n}\widetilde{\mathrm{uSE}}_i\right] = \frac{1}{n}\sum_{i=1}^{n}\mathbb{E}\left[\widetilde{\mathrm{uSE}}_i\right] = \frac{1}{n}\sum_{i=1}^{n}\mathrm{SE}_i = \mathrm{MSE}. \tag{23}$$

### G.2 PROOF OF THEOREM 4

The following lemma bounds the variance of each individual term in the uMSE.

**Lemma 11** (Proof in Section G.5.3). *Under the assumptions of the theorem,*

$$\mathrm{Var}\left[\widetilde{\mathrm{uSE}}_i\right] \leq 14\alpha, \qquad 1 \leq i \leq n. \tag{24}$$

The proof then follows from the fact that the variance of a sum of independent random variables is equal to the sum of their variances,

$$\mathbb{E}\left[\left(\widetilde{\mathrm{uMSE}} - \mathrm{MSE}\right)^2\right] = \mathrm{Var}\left[\widetilde{\mathrm{uMSE}}\right] = \mathrm{Var}\left[\frac{1}{n}\sum_{i=1}^{n}\widetilde{\mathrm{uSE}}_i\right] = \frac{1}{n^2}\sum_{i=1}^{n}\mathrm{Var}\left[\widetilde{\mathrm{uSE}}_i\right] \leq \frac{14\alpha}{n}. \tag{25}$$

The bound immediately implies convergence in mean square as $n \to \infty$, which in turn implies convergence in probability.

### G.3 PROOF OF COROLLARY 5

The uPSNR is a continuous function of the uMSE, which is the same function mapping the MSE to the PSNR. The result then follows from Theorem 4 and the continuous mapping theorem.

### G.4 PROOF OF THEOREM 6

To prove Theorem 6, we express the uMSE as a sum of zero-mean random variables,

$$\widetilde{\mathrm{uMSE}} = \sum_{i=1}^{n}\tilde{t}_i, \qquad \tilde{t}_i := \frac{\widetilde{\mathrm{uSE}}_i - \mathrm{SE}_i}{n}, \tag{26}$$

and apply the following version of the Lyapunov central limit theorem.

**Theorem 12** (Theorem 9.2 Breiman (1992)). *Let $\tilde{t}_i$, $1 \leq i \leq n$, be independent zero-mean random variables with bounded second and third moments, and let*

$$s_n^2 := \sum_{i=1}^n \mathbb{E}\left[\tilde{t}_i^2\right]. \tag{27}$$

*If the Lyapunov condition*

$$\lim_{n \to \infty} \frac{\sum_{i=1}^n \mathbb{E}\left[\left|\tilde{t}_i\right|^3\right]}{s_n^3} = 0 \tag{28}$$

*holds, then the random variable*

$$\frac{1}{s_n} \sum_{i=1}^n \tilde{t}_i \tag{29}$$

*converges in distribution to a standard Gaussian as $n \to \infty$.*

To complete the proof we show that the random variable

$$\tilde{t}_i := \frac{\widetilde{\mathrm{uSE}}_i - \mathrm{SE}_i}{n} \tag{30}$$

satisfies the conditions of Theorem 12. By Lemma 10 its mean is zero. By Lemma 11 its second moment is bounded. To control $s_n$, we apply the following auxiliary lemma, which provides a lower bound on the variance of each term in the uMSE.

**Lemma 13.** *Under the assumptions of Theorem 6,*

$$\mathrm{Var}\left[\widetilde{\mathrm{uSE}}_i\right] \geq \frac{\mu_i^{[4]} + \sigma_i^4}{2}, \tag{31}$$

*where $\mu_i^{[4]}$ and $\sigma_i^2$ denote the fourth central moment and the variance of $\tilde{a}_i$, $\tilde{b}_i$ and $\tilde{c}_i$.*

The lemma yields a lower bound for $s_n^2$,

$$s_n^2 := \sum_{i=1}^n \mathbb{E}\left[\tilde{t}_i^2\right] \tag{32}$$

$$= \frac{1}{n^2} \sum_{i=1}^n \mathbb{E}\left[\left(\widetilde{\mathrm{uSE}}_i - \mathrm{SE}_i\right)^2\right] \tag{33}$$

$$= \frac{1}{n^2} \sum_{i=1}^n \mathrm{Var}\left[\widetilde{\mathrm{uSE}}_i\right] \tag{34}$$

$$\geq \frac{2\mu_i^{[4]} + 2\sigma^4}{n}. \tag{35}$$

The following lemma controls the numerator in the Lyapunov condition, and also shows that the third moment of $\tilde{t}_i$ is bounded.

**Lemma 14** (Proof in Section G.5.5). *Under the assumptions of Theorem 6, there exists a numerical positive constant $D$ such that*

$$\sum_{i=1}^n \mathbb{E}\left[\left|\tilde{t}_i\right|^3\right] \leq \frac{D\eta}{n^2}. \tag{36}$$

Combining equation 35 and Lemma 14, we obtain

$$\frac{\sum_{i=1}^n \mathbb{E}\left[\left|\tilde{t}_i\right|^3\right]}{s_n^3} \leq \frac{D\eta}{(2\mu_i^{[4]} + 2\sigma^4)^{1.5}\sqrt{n}}, \tag{37}$$

which converges to zero as $n \to \infty$. The Lyapunov condition therefore holds and the proof is complete.

## G.5 PROOF OF AUXILIARY RESULTS

### G.5.1 NOTATION

To alleviate notation in our proofs, we define the denoising error $\mathrm{err}_i := f(y)_i - x_i$ and the centered random variables $C(\tilde{a}_i) := \tilde{a}_i - x_i$, $C(\tilde{b}_i) := \tilde{b}_i - x_i$ and $C(\tilde{c}_i) := \tilde{c}_i - x_i$, which are independent, have zero mean and satisfy $\mathrm{Var}\,[\tilde{a}_i] = \mathbb{E}\,[\tilde{a}_i^2] = \mathrm{Var}[\tilde{b}_i] = \mathbb{E}[\tilde{b}_i^2] = \mathrm{Var}\,[\tilde{c}_i] = \mathbb{E}\,[\tilde{c}_i^2]$.

### G.5.2 PROOF OF LEMMA 10

By linearity of expectation and the fact that the variance of independent random variables equals the sum of their variances,

$$\mathbb{E}\left[\widetilde{\mathrm{uSE}}_i\right] = \mathbb{E}\left[(\tilde{a}_i - f(y)_i)^2 - \frac{(\tilde{b}_i - \tilde{c}_i)^2}{2}\right] \tag{38}$$

$$= \mathbb{E}\left[(C(\tilde{a}_i) - \mathrm{err}_i))^2\right] - \frac{\mathbb{E}\left[(C(\tilde{b}_i) - C(\tilde{c}_i))^2\right]}{2} \tag{39}$$

$$= \mathbb{E}\left[C(\tilde{a}_i)^2\right] - 2\mathrm{err}_i\mathbb{E}\left[C(\tilde{a}_i)\right] + \mathrm{SE}_i - \frac{\mathrm{Var}\left[C(\tilde{b}_i) - C(\tilde{c}_i)\right]}{2} \tag{40}$$

$$= \mathrm{Var}\,[\tilde{a}_i] + \mathrm{SE}_i - \frac{\mathrm{Var}[C(\tilde{b}_i)] + \mathrm{Var}\,[C(\tilde{c}_i)]}{2} \tag{41}$$

$$= \mathrm{Var}\,[\tilde{a}_i] + \mathrm{SE}_i - \frac{\mathrm{Var}[\tilde{b}_i] + \mathrm{Var}\,[\tilde{c}_i]}{2} \tag{42}$$

$$= \mathrm{SE}_i. \tag{43}$$

### G.5.3 PROOF OF LEMMA 11

By linearity of expectation, the fact that the variance of independent random variables equals the sum of their variances and the fact that the mean square is an upper bound on the variance,

$$\mathrm{Var}\left[\widetilde{\mathrm{uSE}}_i\right] = \mathrm{Var}\left[(\tilde{a}_i - f(y)_i)^2\right] + \frac{\mathrm{Var}\left[(\tilde{b}_i - \tilde{c}_i)^2\right]}{4} \tag{44}$$

$$\leq \mathbb{E}\left[(\tilde{a}_i - f(y)_i)^4\right] + \frac{\mathbb{E}\left[(\tilde{b}_i - \tilde{c}_i)^4\right]}{4} \tag{45}$$

$$= \mathbb{E}\left[(C(\tilde{a}_i) - \mathrm{err}_i)^4\right] + \frac{\mathbb{E}\left[\left(C(\tilde{b}_i) - C(\tilde{c}_i)\right)^4\right]}{4} \tag{46}$$

$$\leq \mathbb{E}\left[C(\tilde{a}_i)^4\right] + 4\mathbb{E}\left[C(\tilde{a}_i)^3\right]|\mathrm{err}_i| + 6\mathbb{E}\left[C(\tilde{a}_i)^2\right]\mathrm{err}_i^2 + \mathrm{err}_i^4$$

$$+ \frac{\mathbb{E}\left[C(\tilde{b}_i)^4\right] + 6\mathbb{E}\left[C(\tilde{b}_i)^2\right]\mathbb{E}\left[C(\tilde{c}_i)^2\right] + \left[C(\tilde{c}_i)^4\right]}{4} \tag{47}$$

$$\leq 14\alpha, \tag{48}$$

where we have also used the fact that $\mu_2^2 \leq \mu_i^{[4]}$ by Jensen's inequality, which implies

$$\mathbb{E}\left[C(\tilde{a}_i)^2\right]\mathrm{SE}_i \leq \sqrt{\mu_i^{[4]}\gamma^4} \leq \alpha, \tag{49}$$

$$\mathbb{E}\left[C(\tilde{b}_i)^2\right]\mathbb{E}\left[C(\tilde{c}_i)^2\right] \leq \mu_i^{[4]} \leq \alpha. \tag{50}$$

### G.5.4 PROOF OF LEMMA 13

The variance of independent random variables equals the sum of their variances, so

$$\text{Var}\left[\widetilde{\text{uSE}}_i\right] = \text{Var}\left[(\tilde{a}_i - f(y)_i)^2\right] + \frac{\text{Var}\left[(\tilde{b}_i - \tilde{c}_i)^2\right]}{4} \tag{51}$$

$$\geq \frac{\text{Var}\left[(\tilde{b}_i - \tilde{c}_i)^2\right]}{4}. \tag{52}$$

and since the mean of $\tilde{b}_i - \tilde{c}_i$ is zero,

$$\mathbb{E}\left[(\tilde{b}_i - \tilde{c}_i)^2\right] = \text{Var}[\tilde{b}_i - \tilde{c}_i] \tag{53}$$

$$= \text{Var}[\tilde{b}_i] + \text{Var}[\tilde{c}_i] \tag{54}$$

$$= 2\sigma_i^2. \tag{55}$$

By the definition of variance and linearity of expectation,

$$\text{Var}\left[(\tilde{b}_i - \tilde{c}_i)^2\right] = \mathbb{E}\left[(\tilde{b}_i - \tilde{c}_i)^4\right] - \mathbb{E}\left[(\tilde{b}_i - \tilde{c}_i)^2\right]^2 \tag{56}$$

$$= \mathbb{E}\left[\left(C(\tilde{b}_i) - C(\tilde{c}_i)\right)^4\right] - 4\sigma_i^4 \tag{57}$$

$$= \mathbb{E}\left[C(\tilde{b}_i)^4\right] + \mathbb{E}\left[C(\tilde{c}_i)^4\right] + 6\mathbb{E}\left[C(\tilde{b}_i)^2 C(\tilde{c}_i)^2\right] - 4\sigma_i^4 \tag{58}$$

$$= 2\mu_i^{[4]} + 2\sigma_i^4. \tag{59}$$

### G.5.5 PROOF OF LEMMA 14

By linearity of expectation,

$$\mathbb{E}\left[\left|\widetilde{\text{uSE}}_i - \text{SE}_i\right|^3\right] = \mathbb{E}\left[\left|(\tilde{a}_i - f(y)_i)^2 - \frac{(\tilde{b}_i - \tilde{c}_i)^2}{2} - \text{err}_i^2\right|^3\right] \tag{60}$$

$$= \mathbb{E}\left[\left|(C(\tilde{a}_i) - \text{err}_i)^2 - \frac{(C(\tilde{b}_i) - C(\tilde{c}_i))^2}{2} - \text{err}_i^2\right|^3\right] \tag{61}$$

$$\leq \mathbb{E}\left[(C(\tilde{a}_i) - \text{err}_i)^6\right] + \frac{1}{8}\mathbb{E}\left[(C(\tilde{b}_i) - C(\tilde{c}_i))^6\right] + \text{err}_i^6 \tag{62}$$

$$+ \frac{3}{2}\mathbb{E}\left[(C(\tilde{a}_i) - \text{err}_i)^4\right]\mathbb{E}\left[(C(\tilde{b}_i) - C(\tilde{c}_i))^2\right] \tag{63}$$

$$+ \frac{3}{4}\mathbb{E}\left[(C(\tilde{a}_i) - \text{err}_i)^2\right]\mathbb{E}\left[(C(\tilde{b}_i) - C(\tilde{c}_i))^4\right] \tag{64}$$

$$+ 3\mathbb{E}\left[(C(\tilde{a}_i) - \text{err}_i)^4\right]\text{err}_i^2 + 3\mathbb{E}\left[(C(\tilde{a}_i) - \text{err}_i)^2\right]\text{err}_i^4 \tag{65}$$

$$+ \frac{3}{4}\mathbb{E}\left[(C(\tilde{b}_i) - C(\tilde{c}_i))^4\right]\text{err}_i^2 + \frac{3}{2}\mathbb{E}\left[(C(\tilde{b}_i) - C(\tilde{c}_i))^2\right]\text{err}_i^4 \tag{66}$$

$$+ 3\mathbb{E}\left[(C(\tilde{a}_i) - \text{err}_i)^2\right]\mathbb{E}\left[(C(\tilde{b}_i) - C(\tilde{c}_i))^2\right]\text{err}_i^2 \tag{67}$$

$$\leq D\eta. \tag{68}$$

The final bound in equation 68 is obtained by bounding each term in the sum using the assumption that the maximum entrywise denoising error and the central moments of $\tilde{a}_i$, $\tilde{b}_i$ and $\tilde{c}_i$ are bounded. For example,

$$\mathbb{E}\left[(C(\tilde{a}_i) - \text{err}_i)^6\right] = \mathbb{E}\left[C(\tilde{a}_i)^6\right] + \text{err}_i^6 + 6\mathbb{E}\left[C(\tilde{a}_i)^5\right]\text{err}_i + 15\mathbb{E}\left[C(\tilde{a}_i)^4\right]\text{err}_i^2 \tag{69}$$

$$+ 15\mathbb{E}\left[C(\tilde{a}_i)^2\right]\text{err}_i^4 + 20\mathbb{E}\left[C(\tilde{a}_i)^3\right]\text{err}_i^3. \tag{70}$$

Finally, by linearity of expectation we have

$$\sum_{i=1}^{n} \mathbb{E}\left[\left|\tilde{t}_i\right|^3\right] = \frac{1}{n^3} \sum_{i=1}^{n} \mathbb{E}\left[\left|\widetilde{\text{uSE}}_i - \text{SE}_i\right|^3\right] \tag{71}$$

$$\leq \frac{D\eta}{n^2}. \tag{72}$$

## H    DESCRIPTION OF CONTROLLED EXPERIMENTS

In this section, we describe the architectures and training procedure for models used in Section 5. For our experiments with natural images, we use the pre-trained weights released in Zhang et al. (2017a) and Mohan et al. (2020). All models are trained on $180 \times 180$ natural images from the Berkeley Segmentation Dataset (Martin et al., 2001) synthetically corrupted with Gaussian noise with standard deviation uniformly sampled between 0 and 100. The training set contains 400 images and is augmented via downsampling, random flips, and random rotations of patches in these images (Zhang et al., 2017a; Mohan et al., 2020). We use the standard test set containing 68 images for evaluation. We describe each of the models we use in detail below.

1. **Bilateral filter** OpenCV implementation for the Bilateral filter with a filter diameter of 15 pixels and $\sigma_{value} = \sigma_{space} = 1$.

2. **DnCNN**. DnCNN Zhang et al. (2017a) consists of 20 convolutional layers, each consisting of $3 \times 3$ filters and 64 channels, batch normalization (Ioffe & Szegedy, 2015), and a ReLU nonlinearity. It has a skip connection from the initial layer to the final layer, which has no nonlinear units. We use the pre-trained weights released by the authors.

3. **UNet**. Our UNet model (Ronneberger et al., 2015) has the following layers:

   (a) *conv1* - Takes in input image and maps to 32 channels with $5 \times 5$ convolutional kernels.

   (b) *conv2* - Input: 32 channels. Output: 32 channels. $3 \times 3$ convolutional kernels.

   (c) *conv3* - Input: 32 channels. Output: 64 channels. $3 \times 3$ convolutional kernels with stride 2.

   (d) *conv4*- Input: 64 channels. Output: 64 channels. $3 \times 3$ convolutional kernels.

   (e) *conv5*- Input: 64 channels. Output: 64 channels. $3 \times 3$ convolutional kernels with dilation factor of 2.

   (f) *conv6*- Input: 64 channels. Output: 64 channels. $3 \times 3$ convolutional kernels with dilation factor of 4.

   (g) *conv7*- Transpose Convolution layer. Input: 64 channels. Output: 64 channels. $4 \times 4$ filters with stride 2.

   (h) *conv8*- Input: 96 channels. Output: 64 channels. $3 \times 3$ convolutional kernels. The input to this layer is the concatenation of the outputs of layer *conv7* and *conv2*.

   (i) *conv9*- Input: 32 channels. Output: 1 channels. $5 \times 5$ convolutional kernels.

   We use pre-trained weights released by the authors of Mohan et al. (2020).

4. **DenseNet** The simplified version of the DenseNet architecture (Huang et al., 2017) has 4 blocks in total. Each block is a fully convolutional 5-layer CNN with $3 \times 3$ filters and 64 channels in the intermediate layers with ReLU nonlinearity. The first three blocks have an output layer with 64 channels, while the last block has an output layer with only one channel. The output of the $i^{th}$ block is concatenated with the input noisy image and then fed to the $(i + 1)^{th}$ block, so the last three blocks have 65 input channels. We use pre-trained weights released by the authors of Mohan et al. (2020).

For our experiments with electron microscopy data, we use the simulated dataset of Pt nanoparticles introduced in Mohan et al. (2022). Specifically, we used a subset of 5,583 images corresponding to white contrast (the simulated dataset is divided into white, black and intermediate contrast by a domain expert, see Mohan et al. (2022) for more details). 90% of the data were used for training. The remaining 559 images were evenly split into validation and test sets. The UNet architecture used in these experiments is the one introduced in Mohan et al. (2022) with 4 scales and 32 base channels. In addition to bilateral filter, UNet, and DnCNN models described for natural images, we used a

blindspot based network. BlindSpot Laine et al. (2019) is a CNN which is constrained to predict the intensity of a pixel as a function of the noisy pixels in its neighbourhood, without using the pixel itself. Following Laine et al. (2019); Sheth et al. (2021), we use a UNet architecture as the model backbone.

## I    DESCRIPTION OF EXPERIMENTS WITH VIDEOS IN RAW FORMAT

As explained in Section 6, the dataset contains 11 unique videos, each containing 7 frames, captured at five different ISO levels using a surveillance camera. Each video has 10 different noise realizations per frame, which are averaged to obtain an estimated clean version of the video. Following Sheth et al. (2021), we perform evaluation on five videos from the test test.

The methods we use:

1. **Image Denoiser (Wavelet)**. We use Daubechies wavelet to perform denoising, which is the default choice in *skimage.restoration*, a widely used image restoration package. We implement denoising using the function *denoise_wavelet()* from the package using the default options. We set *sigma=0.01*.

2. **Image Denoiser (CNN)**. We perform image denoising by re-purposing the video denoiser (UDVD) trained for RAW videos in Ref. Sheth et al. (2021). UDVD takes in five consecutive frames, and output the denoised image corresponding to the frame in the middle. To simulate image denoising using UDVD, we repeat the same frame 5 times (i.e, all frames are the same image), and provide it as input to the trained network.

3. **Video Denoiser (Temp. Avg.)**. We use 5 consecutive frames to compute the denosied image corresponding to the middle frame. We assign a weight of $0.75$ to the middle noisy frame, $0.1$ to each of the previous and next frame, and $0.025$ to the rest of the two frames.

4. **Video Denoiser (CNN)**. We use the unsupervised video denoiser (UDVD) trained for RAW videos in Ref. Sheth et al. (2021). As explained, UDVD takes in five consecutive frames, and output the denoised image corresponding to the frame in the middle. We use the pre-trained weights, and follow the experimental setup described in Ref. Sheth et al. (2021).

We use the pre-trained weights released by the authors of Sheth et al. (2021) as our image and video denoiser. These weights are obtained by training UDVD on the first 9 realizations of the 5 videos from the test set of the raw video dataset, holding out the last realization for early stopping (see Sheth et al. (2021) for more details).

## J    DESCRIPTION OF EXPERIMENTS ON ELECTRON MICROSCOPY DATA

**Data acquisition:**

The dataset contains TEM images of Pt nanoparticles on a CeO2 substrate. An electron beam interacts with the sample, and then its intensity is recorded on an imaging plane by the detector. The pixel intensity approximately follows a Poisson distribution with parameter equal to the intensity of the electron beam.

The data were recorded at room temperature at a pressure of $\sim 10 - 6$ Torr. The electron beam intensity was $600e/\text{Å}^2/s$. The instances are part of 25 videos, taken at a frame rate of 75 frames per second. The instances show Pt particles in the size range 1 - 5 nm. In a subset of frame series, the particles become unstable and undergo structural dynamic re-arrangements. The periods of instability are punctuated by periods of relative stability. Consequently, the nanoparticles show a variety of different sizes and shapes and are also viewed along many different crystallographic directions. Data were collected using a FEI Titan ETEM in EFTEM mode, Gatan Tantalum hot stage, K3 camera in CDS counting mode.

**Pixel-wise correlation:** Our proposed unsupervised metrics rely on the assumption that the noise is pixel-wise independence. This is not the case for this dataset, as shown in Figure 13.We address this by performing spatial subsampling by a factor of two, which reduces the pixel-wise correlation by an order of magnitude. After this, some frames still present relatively high pixel-wise correlations. We therefore select the test sets from two sets of contiguous frames with low correlation.

**Training and test sets:** The data were divided into three sets: training & validation set, consisting of 70% of the data, and two contiguous test sets with pixel-wise correlation: one containing 155 images with *moderate* signal-to-noise ratio (SNR), and one containing 383 images with *low* SNR, which are more challenging. The moderate SNR test set is interspersed with the training and validation sets, and contains frames similar to those used to train the network. The low SNR test set contains frames which are temporally separated from the training and validation sets, and contains nanoparticle with different structures.

**Denoisers** We compare the performance of two denoisers: (1) a convolutional neural network based on Neighbor2Neighbor with the same architecture as in Huang et al. (2021); (2) Gaussian smoothing with kernel size =25 and $\sigma = 25$.

**CNN training parameters** The CNN was based on the architecture in Huang et al. (2021). The network consists of a modified UNet, and is trained for 500 epochs using an Adam optimizer with an initial learning rate of 0.0001, and scheduled reduction of the learning rate every 100 epochs. The network has a total of 1,256,689 parameters.

