# OpenReview forum: "Evaluating Unsupervised Denoising Requires Unsupervised Metrics"
_ICLR.cc/2023/Conference — Submitted to ICLR 2023_

### Official Review · Reviewer_NUcX · 2022-10-24

**Confidence:** 4
**Correctness:** 4
**Technical Novelty And Significance:** 2
**Empirical Novelty And Significance:** 3
**Recommendation:** 6

**Clarity, Quality, Novelty And Reproducibility:**

The paper is well written and the exposition is in general good. The analysis of the related work is a little light and in order to clearly judge the contribution this needs to be improved. The novelty seems limited, given that the approach seems a straightforward extension of noise2noise with a noise estimation (using two additional noisy frames).

**Strength And Weaknesses:**

Strengths of the paper:
1.  Tackles a very relevant problem (both practical and theoretical).
2. Presents a sound solution with very clear requirements, and later introduces two ideas on how to approximately meet the requirements so we can get approximately unbiased estimator of MSE/PSNR.
3. Shows results on real images (electron microscopy) as long as synthetic cases, and also introduces a nice discussion about the limitations of the method.

Weakness:

1. No discussion regarding a very close and relevant problem on Computer Graphics – the problem of Monte Carlo rendering. Where for getting an estimate of MSE, the method keeps track of independent buffers and combines the estimates. See for example this overview paper and the references therein:

* Zwicker, M., Jarosz, W., Lehtinen, J., Moon, B., Ramamoorthi, R., Rousselle, F., Sen, P., Soler, C. and Yoon, S.E., 2015, May. Recent advances in adaptive sampling and reconstruction for Monte Carlo rendering. In Computer graphics forum (Vol. 34, No. 2, pp. 667-681).

2. No discussion regarding similar methods but that rely on the Stein Unbiased Risk Estimator (SURE) (Just a minor mention to this under the paragraph "Unsupervised Denoising"). The paper would be much stronger with a more detailed comparison (technical, not necessarily empirical) to for example these works:
* Soltanayev, S. and Chun, S.Y., 2018. Training deep learning based denoisers without ground truth data. Advances in neural information processing systems, 31.
* Zhussip, M., Soltanayev, S. and Chun, S.Y., 2019. Extending stein's unbiased risk estimator to train deep denoisers with correlated pairs of noisy images. Advances in neural information processing systems, 32.

3. No discussion about the dual problem of noise (level) estimation. Tone of literature on image denoising (from image processing) and noise level estimation. See for example:
* Liu, X., Tanaka, M. and Okutomi, M., 2013. Single-image noise level estimation for blind denoising. IEEE transactions on image processing, 22(12), pp.5226-5237.
* Lebrun, M., Colom, M. and Morel, J.M., 2015. Multiscale image blind denoising. IEEE Transactions on Image Processing, 24(10), pp.3149-3161.
* Arias, P. and Morel, J.M., 2018. Video denoising via empirical Bayesian estimation of space-time patches. Journal of Mathematical Imaging and Vision, 60(1), pp.70-93.

Also in all the shown cases, the model of the noise is Poisson or Gaussian. This information about the noise model  could be used explicitly so the problem becomes much easier (see, e.g, Prakash [2021]). The paper will be much stronger if the examples and the formulation is given for cases where the noise model is not known or it's not easy to model (mix of things).

Prakash, M., Delbracio, M., Milanfar, P. and Jug, F., 2021, September. Interpretable Unsupervised Diversity Denoising and Artefact Removal. In International Conference on Learning Representations.


**Summary Of The Paper:**

This paper proposes a method to estimate the MSE (and PSNR) in image denoising using only noisy images (no reference). The main idea is to use three noisy images to produce an unbiased estimator of the MSE. The method is sound and well motivated. Several results on synthetic cases and a real example on denoising Electron Microscopy is presented.

**Summary Of The Review:**

I think this is a good paper with an interesting idea. The main weakness in the current exposition is the lack of discussion with respect to other existing work. It is hard to judge the contributions (in particular the novelty of the method). I'm happy to update my review if the authors provide a more detail comparison and analysis of the raised points.

After Rebuttal.
The authors addressed the concerns raised on the first round of reviews and updated the manuscript accordingly. I'm leaning towards accepting this paper, and I'm updating my score to accordingly.

---

> ### Author Response · Authors · 2022-11-18
> **Response to reviewer NUcX**
>
>
> We thank the reviewer for their thoughtful feedback, which we believe has helped us to improve the paper. We completely agree that the related work was incomplete and we have extended it substantially. We believe that this better sets the context for our contributions: two metrics for unsupervised evaluation which (1) are novel, (2) have theoretical guarantees, (3)  are shown to be effective via controlled numerical experiments with synthetic noise, (4) are validated on real-world data from two imaging modalities.
>
> In addition, we have realized that we had not explained the requirements of the proposed metrics and our experiments with real data sufficiently clearly.  The metrics do not assume a specific noise model. Our controlled experiments are carried out with Gaussian and Poisson noise, but this is not used explicitly when computing the metric. In addition, we do have two examples where the noise is unknown: the two real-world data applications in Sections 6 and 7. In both cases, the noise model corrupting the images is unknown. In the case of the raw videos in Section 6, the noise model actually conforms to the reviewer’s suggestion: it is an unknown combination of shot noise and read noise as explained in Yue et al 2020. We have edited the introduction to make this clearer. We also cite Prakash et al 2020 in Section 8.
>
> We address each of the reviewer’s comments in more detail below:
>
> 1. We now mention the related problem of realistic image synthesis in Section 4.
>
> 2. We fully agree with the reviewer that a more detailed discussion on Stein’s unbiased risk estimator (SURE) is warranted. We have therefore added a subsection on SURE in Section 2. SURE provides an asymptotically unbiased estimator of the MSE for i.i.d. Gaussian noise, and can be adapted to other noise distributions. This cost function has been successfully used for training unsupervised denoisers based on deep learning. In principle, SURE could be used to compute the MSE for evaluation, but it has certain limitations: (1) a closed form expression of the noise likelihood is required, including the value of the noise parameters (for example, this is not known for the two real-world datasets in our work, (2) computing SURE requires approximating the divergence of a denoiser (usually via Monte Carlo methods), which is computationally very expensive. That said, developing practical unsupervised metrics based on SURE and studying their theoretical properties is definitely an interesting direction for future research.
>
> 3. We agree with the reviewer that noise-level estimation is related to our proposed metrics. The correction term in uMSE can be interpreted as an estimate of the noise level, obtained by cancelling out the clean signal. In this sense, it is related to noise-level estimation methods. However, unlike uMSE, these methods typically assume a parametric model for the noise, and are not used for evaluation. We have added a subsection on noise-level estimation methods in Section 2 to clarify this.

---

### Official Review · Reviewer_wQ5W · 2022-10-25

**Confidence:** 4
**Clarity, Quality, Novelty And Reproducibility:** The paper is easy to read.
**Correctness:** 4
**Technical Novelty And Significance:** 3
**Empirical Novelty And Significance:** 2
**Recommendation:** 5

**Strength And Weaknesses:**

Strength:

1. The idea of using multiple noisy images to construct unsupervised metrics is interesting and novel.

2. The performance of the proposed methods uMSE and uPSNR is comparable with the supervised metrics MSE and PSNR.

3. The author provided the statistical analysis of the proposed method under some ideal assumptions.

Weakness:

1. This method's key point is using three noisy references to compute the unsupervised metrics. However, they used existing methods (consecutive frames and Neighbor2Neighbor (CVPR2021)) to construct those three reference images. It should be discussed for clarifying the significance of the paper.

2. The independence assumption made in the theoretical analysis section does not hold in practice. The methods they used cannot produce perfect noisy references which satisfy those assumptions since there will be misalignment in consecutive frames and subsampling images.In this sense, it should discuss the case if the noisy references have some errors.

3. There are other no-referenced image quality assessment methods, such as Natural Image Quality Evaluator (NIQE), and BRISQUE [1]; the author should discuss the difference between those methods.

4. This method only works for noise distributions that are pixel-wise independent, which limits its applications.

5. The analysis shows that it provides an unbiased estimator, but the variance of the estimator should be discussed in the main text.

[1] Mittal A, Moorthy A K, Bovik A C. No-reference image quality assessment in the spatial domain[J]. IEEE Transactions on image processing, 2012, 21(12): 4695-4708.
[2] Claude E Shannon. A mathematical theory of communication. The Bell system technical journal, 27(3):379–423, 1948.





**Summary Of The Paper:**

This paper proposed two unsupervised metrics for evaluating image qualities: uMSE and uPSNR. The main idea is to use three noisy references to design an estimator for MSE, which is the uMSE. The noisy references can be computed using consecutive frames or applying spatial subsampling. It uses the uMSE to define the unsupervised PSNR (uPSNR) metric and demonstrates that the proposed metrics are unbiased and consistent estimators of the supervised metrics under the independence and centered noise assumptions.

**Summary Of The Review:**

The paper considers the image quality assessment problem which is important for the applications. But, the author should address the mentioned concerns that show the advantages of the proposed method.

---

> ### Author Response · Authors · 2022-11-18
> **Response to reviewer wQ5W**
>
> We thank the reviewer for their comments, which we believe have helped us improve the clarity of the manuscript. We respond to each of them in detail below:
>
> 1. Our key contribution with respect to the existing unsupervised denoising methods is a novel unsupervised metric that can be used for evaluation, as it is designed to be an unbiased and consistent estimator of the MSE, unlike the existing cost functions used in Noise2Noise and Neighbor2Neighbor. We provide theoretical guarantees, show that it provides accurate evaluation via controlled numerical experiments with synthetic noise, and validate it on real-world data from two imaging modalities. We have edited Section 2 to clarify this.
>
> 2. In some applications, the independence assumption is in fact realistic. This is the case for the raw video application in Section 6 and for the electron microscopy application in Section 7. Figure 13 provides a graph of the pixel-wise correlation in the electron microscopy data. After subsampling once to remove the correlation between adjacent pixels, the correlation coefficient is 0.001. That said, we agree with the reviewer that accounting for noise that is not pixel-wise independent would be very interest, but we do want to emphasize that most current unsupervised deep-learning denoisers assume independent noise (see below).
>
> 3. We have added a discussion on no reference image quality assessment methods in Section 2. We agree with the reviewer that the difference of our approach and these methods should be discussed.
>
> 4. While we agree that dealing with noise that is not pixel-wise independent would be of interest (as we point out in Section 8), current unsupervised deep learning methods such as Noise2Noise, Noise2Self, Neighbor2Neighbor, etc. are all designed for pixel-wise independent noise. It is therefore of interest to be able to evaluate their performance in an unsupervised fashion under this assumption, which can be done with the proposed metrics. In addition, our results on real-world data from two different imaging modalities shows that this assumption does hold approximately in certain practical applications.
>
> 5. We have edited Theorem 4 to make it clear that it provides a bound on the variance of the estimator.

---

### Official Review · Reviewer_zhXu · 2022-10-25

**Confidence:** 4
**Clarity, Quality, Novelty And Reproducibility:** The work is clearly written, addressi…
**Correctness:** 4
**Technical Novelty And Significance:** 3
**Empirical Novelty And Significance:** 3
**Recommendation:** 6

**Strength And Weaknesses:**

The paper is clearly written and clearly states the contributions and limitations of the method, experiments and results.

Questions:
-  Table 1: the authors point that the metrics are less accurate on the natural image dataset when SNR is high given the bias introduced by susbsampling, but ok on the TEM dataset which is smoother and has high spatial resolution. Given that this is a methods paper, i think the paper would reach a wider audience from a more detailed discussion on how this finding should affect the use of the metric. For example, if my dataset has temporal resolution and we have to use subsampling to get the noisy estimates to calculate uMSE, should I use MSE_avg (standard approach)?
- What is the conclusion in Fig 6? Can we say that when uMSE is biased it underestimates the true MSE. Do the authors expect the method to always underestimate it or could it overestimate it as well? Having some additional dataset would be helpful here.
- In Fig 10, the distributions of UMSE are very similar, do the authors think that this holds on a per data point level and the uMSE can help determine if different methods identify the same images as noisy? For example, one could produce scatter plots of the uMSE for different approaches (each in x,y axes) for individual data points to determine the correlation.


**Summary Of The Paper:**

This paper proposes two metrics for unsupervised error/noise estimation, the methods are unsupervised because they can be calculated using noisy images as opposed to clean/noisy images. The metrics are a uMSE and uPSNR (based on uMSE), and the authors show that the consistency of the metrics, and state several limitations of the proposed approaches.


**Summary Of The Review:**

Overall, this paper proposes two metrics for unsupervised error/noise estimation. The paper is clearly written and addresses a problem which is typically overlooked. I would like to see this work published although the contributions of this work are fairly limited and it might be better suited for a workshop.

---

> ### Author Response · Authors · 2022-11-18
> **Response to reviewer zhXu**
>
> We thank the reviewer for the comments and the useful feedback. We have edited the paper to address their questions, incorporating their suggested plot that demonstrates that the proposed metrics identify noisy reconstructions in the real data (see below). Regarding our contributions, we propose two metrics for unsupervised evaluation which (1) are novel, (2) have theoretical guarantees, (3)  are shown to be effective via controlled numerical experiments with synthetic noise, (4) are validated on real-world data from two imaging modalities. We believe that these contributions warrant publication in ICLR.
>
> - Question 1: We agree with the author that a discussion of how to compute the noisy references in practice would strengthen the paper. We have added it in Section 3.3. We have also added a controlled experiment in Section D that shows that spatial subsampling is effective as long as the image content is sufficiently smooth with respect to the pixel resolution.
>
> - Question 2: Figure 6 shows that when the noisy references correspond to the same clean underlying image, the confidence intervals in Section B are valid confidence intervals, but this may not be the case when spatial subsampling is applied if the underlying image content is not sufficiently smooth with respect to the image resolution. We have edited the caption to clarify this. The uMSE does not always underestimate the MSE, as shown in the right histogram of Figure 3.
>
> - Question 3: We thank the reviewer for the great suggestion! We have added the suggested scatterplot as Figure 12. While the ranges are different for each denoiser (the CNN denoises more effectively than Gaussian smoothing), the uMSE values of the images are highly correlated, indicating that uMSE provides a consistent evaluation at the level of individual images.

---

### Official Review · Reviewer_Y2NW · 2022-10-28

**Confidence:** 5
**Correctness:** 3
**Technical Novelty And Significance:** 3
**Empirical Novelty And Significance:** 3
**Recommendation:** 5

**Clarity, Quality, Novelty And Reproducibility:**

The manuscript is presenting a novel and very relevant idea in a very accessible and clear way.
Reproducibility is not a problem at all.

There is one aspect that I find too little developed, namely the limitations and evaluation divergence of the single-image (spatially subsampled) application of the proposed metrics uMSE_s and uPSNR_s. Here something the authors either chose to not discuss or, maybe, have themselves not yet thought through:
* Ripping a single image apart into multiple images so that the original uMSE and uPSNR metrics can be computed is biasing the evaluation heavily against images that contain very highly spatial signal frequencies. (Or the other way around: smooth images will lead to better numbers.)
* This exact point is, I presume, also the reason why natural images lead to the reported bias, while EM images don’t. Microscopy data typically has a pixel resolution that is higher than the “optical” resolution, meaning that the signal is relatively speaking smooth and pixels sample those structures typically above Nyquist.
* The authors mention “are smoother and have higher spatial resolution” which is quite confusing. EM images are smoother (see above), but the spatial resolution is completely irrelevant for the observed effect. I am quite certain that the important term is the difference between the pixel and optical resolution. Making this point crystal clear would improve the paper considerably and I suggest the authors to consider an overhaul of these aspects in their submitted manuscript.

**Strength And Weaknesses:**

*Strength:*
1. The biggest strength of this paper is the idea, which indeed addresses an important and open problem in a flourishing community.
2. The idea is simple, which is a good thing, and the presentation of it excellent.
3. The manuscript also contributes interesting and relevant theorems and proofs that, for example, show that the proposed method converges to the right evaluation.
4. The additional idea of spatial subsampling is an additional contribution of high practical relevance (while at the same time being less clean and leading to some issues, as pointed out below).

*Weaknesses:*
1. While the ideas and results are very clean for the idea presented e.g. in Figure 1 (3-image metrics), the single image application via spatial subsampling is less clean and seems less thoroughly understood by the authors (at least it is, in my point of view, not sufficiently well described). For example, the fact that the computed numbers are biased for natural images but not for EM images is presented without sufficiently explaining why this effect happens for one but not the other modality.
2. I would have wished that the original 3-image metrics would be better separated from the 1-image (spatially subsampled) ideas and results. It is of course distinguished in main text tables and figures, but while reading I still needed multiple times to sort my mind to sort my growing understanding correctly.
3. I missed a more detailed discussion about what noises can be evaluated by the proposed methods. All noises that are independent per pixel given the signal (e.g. Gaussian and Poisson noise), or also structured noises (where neighboring pixels can be affected in correlated ways (e.g. streaking artifacts in tomography)?.

**Summary Of The Paper:**

The authors propose uMSE and uPSNR, two unsupervised metrics to evaluate the quality of denoising methods. The big thrust of the paper stems from the obvious observation that unsupervised (or self-supervised) denoising methods cannot otherwise be evaluated.
This is, in fact, a true limitations in the field for several years now, and the community is so far evaluating unsupervised and self-supervised denoising methods on either artificially noised image data or on data where a good approximation of the ground-truth true signal can be recorded.
Still, the reason why self-supervised denoising methods celebrate the success they have is that they can be applied to data for which the ground truth signal cannot (easily) be recorded and is therefore typically not available.
The authors show that their fundamental idea (requiring 3 independently noisy measurements of the same GT image (signal)) is asymptotically consistent with the supervised MSE and PSNR metrics.
Additional to this idea, the authors also present a variation of their metrics that can be computed from single noisy measurements (images) by first applying a specific spatial subsampling scheme.

**Summary Of The Review:**

Love the idea, like the paper, pointed to the one place I believe should change before publication. Changes should be doable since they would only be require modifications in how the single-image metrics are discussed.

I am sure that many people will start to use the proposed metrics in their future denoising works (me included).

I chose to select “6: marginally above the acceptance threshold“, but depending on the rebuttal conversations I am absolutely willing to change this opinion to a better rating.

---

> ### Author Response · Authors · 2022-11-18
> **Response to reviewer Y2NW**
>
> We thank the reviewer for the encouraging comments and the useful feedback. We completely agree that further elaborating on the single-image application strengthens the paper, and we have modified the paper to address this, as well as the other comments. In more detail:
>
> - To address Weakness 1, we have added a section in the appendix (Section D: Effect of Spatial Subsampling on the Proposed Metrics) which studies the effect of spatial subsampling on natural images. In this section, we report a controlled experiment that shows that the key property is smoothness of the image with respect to the pixel resolution, as pointed out by the reviewer (see Figure 9). We have edited our explanation of the reason behind the bias caused by spatial subsampling accordingly. We completely agree with the reviewer that the previous phrasing was confusing.
>
> - To address Weakness 2, we have changed the main text to indicate the single-image approach and results more clearly. In particular, we have (1) modified Section 3.3, adding a subsection entitled “Single image”, and (2) modified Section 5, adding a subsection entitled “Single-image results”.
>
> - To address Weakness 3, we have restructured Section 4 in order to indicate the assumptions on the noise model, and separate them clearly from the theoretical guarantees.

---

> > ### Comment · Reviewer_Y2NW · 2022-11-18
> > **Followup question...**
> >
> > Thanks for your feedback.
> >
> > Can you please point me to the changes in the manuscript that address the following of my original points raised?
> >
> > * There is one aspect that I find too little developed, namely the limitations and evaluation divergence of the single-image (spatially subsampled) application of the proposed metrics uMSE_s and uPSNR_s. Here something the authors either chose to not discuss or, maybe, have themselves not yet thought through:
> >
> >    * Ripping a single image apart into multiple images so that the original uMSE and uPSNR metrics can be computed is biasing the evaluation heavily against images that contain very highly spatial signal frequencies. (Or the other way around: smooth images will lead to better numbers.)
> >    * This exact point is, I presume, also the reason why natural images lead to the reported bias, while EM images don’t. Microscopy data typically has a pixel resolution that is higher than the “optical” resolution, meaning that the signal is relatively speaking smooth and pixels sample those structures typically above Nyquist.
> >    * The authors mention “are smoother and have higher spatial resolution” which is quite confusing. EM images are smoother (see above), but the spatial resolution is completely irrelevant for the observed effect. I am quite certain that the important term is the difference between the pixel and optical resolution. Making this point crystal clear would improve the paper considerably and I suggest the authors to consider an overhaul of these aspects in their submitted manuscript.
> >
> > Thanks!

---

> > > ### Author Response · Authors · 2022-11-18
> > > **Response to follow up**
> > >
> > > Here is a more detailed account of how we have addressed the divergence between uMSE and uMSE_s. If we have misunderstood any of the points please let us know. Overall, we have:
> > >
> > > 1. Included a figure showing the bias introduced by uMSE_s for a natural image and an electron microscope image (Figure 3). (This was included in the original manuscript)
> > > 2. Included a figure that shows the difference between the underlying subsampled images for natural images and electron microscope images (Figure 8). (This was included in the original manuscript)
> > > 3. Inspired by the comments of the reviewer, we have performed an additional controlled experiment where we applied different degrees of smoothing (via a Gaussian filter) to a natural image. We evaluated the relative RMSE of the corresponding subsampled references. In addition, we fed the smoothed images contaminated by noise into a denoiser and compared the uMSE of the denoised image with its true MSE. The results are shown in Figure 9. We observe that smoothing results in a stark decrease of both the relative RMSE and the uMSE, suggesting that spatial subsampling is effective as long as the underlying image content is sufficiently smooth with respect to the pixel resolution (as supported also by the results on the electron-microscopy data). This is explained in a new section of the appendix (Section D: Effect of Spatial Subsampling on the Proposed Metrics) and a new figure (Figure 9).
> > >
> > > See below for a response point by point.
> > >
> > > **There is one aspect that I find too little developed, namely the limitations and evaluation divergence of the single-image (spatially subsampled) application of the proposed metrics uMSE_s and uPSNR_s. Here something the authors either chose to not discuss or, maybe, have themselves not yet thought through:**
> > > - **Ripping a single image apart into multiple images so that the original uMSE and uPSNR metrics can be computed is biasing the evaluation heavily against images that contain very highly spatial signal frequencies. (Or the other way around: smooth images will lead to better numbers.)**
> > >
> > > We believe this is demonstrated by Table 1, Figure 3, Figure 8 and Figure 9. We comment on this in the “Single-image results” part of Section 5, we mention it as a limitation in Section 8, and we explain it more in detail in the new Section D.
> > >
> > > - **This exact point is, I presume, also the reason why natural images lead to the reported bias, while EM images don’t. Microscopy data typically has a pixel resolution that is higher than the “optical” resolution, meaning that the signal is relatively speaking smooth and pixels sample those structures typically above Nyquist.**
> > >
> > > We fully agree with the reviewer. This comment motivated our additional controlled experiment where we applied different degrees of smoothing (via a Gaussian filter) to a natural image. The results, shown in Figure 9, demonstrate that image content that is smoother with respect to the pixel resolution (which is fixed in the controlled experiment), leads to a sharp decrease in the relative RMSE between subsampled references, and that this is reflected in a decreasing absolute difference between the MSE and the uMSE. This is explained in the new section (Section D: Effect of Spatial Subsampling on the Proposed Metrics).
> > >
> > > - **The authors mention “are smoother and have higher spatial resolution” which is quite confusing. EM images are smoother (see above), but the spatial resolution is completely irrelevant for the observed effect. I am quite certain that the important term is the difference between the pixel and optical resolution. Making this point crystal clear would improve the paper considerably and I suggest the authors to consider an overhaul of these aspects in their submitted manuscript.**
> > >
> > > We completely agree with the reviewer. We have removed the reference to “higher spatial resolution”, and have now written: “this approach is effective as long as the image content is sufficiently smooth with respect to the pixel resolution” (Section 3.3, which we have now restructured to distinguish the single-image approach clearly), “the difference is more pronounced in natural images than in electron-microscopy images, which are smoother with respect to the pixel resolution” (Section 5, in the new subsection “Single-image results”). We also mention this in the captions of Figure 6 and 8, and then explain it more in detail in the new Section D.

---

### Decision · Program_Chairs · 2023-01-20

**Decision:**

Reject

**Justification For Why Not Higher Score:**

For a setup where the uMSE is estimated based on a single image, the uMSE is not reliable, and should not be used for evaluation. This limitation is not sufficiently clear from the paper, and this significantly limits the applicability of the method.


**Justification For Why Not Lower Score:**

N/A

**Metareview: Summary, Strengths And Weaknesses:**

Unsupervised denoising is an important problem setup, since in many real-world denoising applications only noisy images are available without corresponding clean ground-truth images. A variety of methods are available for training networks in an unsupervised manner, and for reconstructing an image based on noisy images alone. However, no metrics are available to evaluate such unsupervised methods based on noisy images alone.

The paper under review proposes an estimator for the MSE (and PSNR) between an estimate of a clean image based on a noisy image, and a clean image that can be computed from three independent noisy observations of the clean image. This metric is called unsupervised MSE (uMSE) and unsupervised PSNR (uPSNR). The idea is a relatively straightforward extension of the noise2noise unsupervised loss. Specifically, the noise2noise loss already provides a biased estimate of the MSE based on two independent noisy versions of a clean image, and the third noisy copy is used to estimate the bias. Subtracting the estimate of the bias yields an unbiased estimate of the MSE.

The paper received four reviews, and a subset of the reviewers and the AC discussed the paper in a virtual meeting. The reviewers are split regarding whether the paper falls slightly below or above the acceptance threshold; two rate the paper slightly below the threshold, and one of the reviewers rating it above the threshold notes that 'I would like to see this work published although the contributions of this work are fairly limited'.

All reviewers and I agree that the paper tackles a relevant problem and that the paper is well-written and the claims are sound. The paper was discussed in a virtual meeting.

The main weakness of the paper is the following:

For the setup where three independent noisy copies of a clean image can be obtained, and where the noise is pixel-wise independent, the proposed metric uMSE is an unbiased estimate of the MSE, as desired. In order to use this estimate we have to be sure that those conditions are satisfied. This limits the applicability to a very particular setup, but as pointed out by R1, an expert in the area, there are indeed setups in microscopy for which three independent noisy copies can be measured and for which the proposed metric makes sense.

However, the paper also proposes to compute the uMSE based on a single noisy image, by splitting a single noisy image into several parts, following ideas from neighbor2neighbor and related methods. Using this approach for obtaining three noisy copies, the uMSE is not an unbiased estimate of the MSE anymore, and becomes unreliable. This is also what the results in the paper show, for the EM dataset the uMSE is a good estimate of the MSE, but for the natural-image dataset it is not.

Thus, for a setup where the uMSE is estimated based on a single image, the uMSE is not reliable, and should not be used for evaluation. This limitation is not sufficiently clear from the paper.


**Summary Of Ac-Reviewer Meeting:**

In a virtual meeting, the consensus emerged that the paper's biggest weakness is that if an estimate of the MSE is obtained based on a single image, it is unreliable. This is not sufficiently clarified in the paper, as explained in the meta-review.

All reviewers found the paper to be very well written, and the problem setup to be interesting.
R2 noted that they would like to see this work published although the contributions of this work are fairly limited and it might be better suited for a workshop.'; I agree that the paper's idea is straightforward given the noise2noise paper, and since the metric is not widely applicable, the contribution is somewhat limited, at least in its current form.

The main reason for recommending to reject at this point is however that the limitation of the metric, in particular when computing uMSE from a single noisy image is not sufficiently well explained. This was pointed out by R1 and discussed in the virtual meeting.